# Light-MILPopt: Solving Large-scale Mixed Integer Linear Programs with Lightweight Optimizer and Small-scale Training Dataset

**Huigen Ye,   Hua Xu** [*]  **Hongyan Wang**
State Key Laboratory of Intelligent Technology and Systems, Department of Computer Science and Technology, Tsinghua University, Beijing 100084, China
Beijing National Research Center for Information Science and Technology(BNRist), Beijing 100084, China
`{yhg23, why17}@mails.tsinghua.edu.cn, xuhua@tsinghua.edu.cn`

## Abstract

Machine Learning (ML)-based optimization approaches emerge as a promising technique for solving large-scale Mixed Integer Linear Programs (MILPs). However, existing ML-based frameworks suffer from high model computation complexity, weak problem reduction, and reliance on large-scale optimizers and large training datasets, resulting in performance bottlenecks for large-scale MILPs. This paper proposes Light-MILPopt, a lightweight large-scale optimization framework that only uses a lightweight optimizer and small training dataset to solve large-scale MILPs. Specifically, Light-MILPopt can be divided into four stages: Problem Formulation for problem division to reduce model computational costs, Model-based Initial Solution Prediction for predicting and constructing the initial solution using a small-scale training dataset, Problem Reduction for both variable and constraint reduction, and Data-driven Optimization for current solution improvement employing a lightweight optimizer. Experimental evaluations on four large-scale benchmark MILPs and a real-world case study demonstrate that Light-MILPopt, leveraging a lightweight optimizer and small training dataset, outperforms the state-of-the-art ML-based optimization framework and advanced large-scale solvers (e.g. Gurobi, SCIP). The results and further analyses substantiate the ML-based framework's feasibility and effectiveness in solving large-scale MILPs.

## 1 Introduction

Mixed Integer Linear Programs (MILPs) are linear optimization problems in which some or all decision variables are subject to integer constraints (Wolsey, 2020). They are widely utilized for modeling combinatorial optimization problems such as routing (Li et al., 2018), bin packing (Paquay et al., 2016) and resource allocation (Kotb et al., 2016). In many scenarios, a large number of homogeneous MILPs with similar combinatorial structures need to be solved simultaneously. In such cases, Machine Learning (ML)-based optimization frameworks can explore correlations between the structure and solution values to enhance solving performance, making ML-based frameworks a promising direction (Han et al., 2023; Deb et al., 2023).

As the pioneering work using ML-based frameworks to solve MILPs, Gasse et al. (2019) proposed a lossless graph representation of MILPs using bipartite graphs, and further accelerated MILP solving by using a Graph Neural Network (GNN) model to learn variable selection policies within the branch-and-bound method. With the increase in the dimensionality of decision variables in MILPs, the search space for the branch-and-bound algorithm expands exponentially. This results in prohibitively high computational costs, especially in the case of large-scale MILPs. To mitigate this challenge, Nair et al. (2020) proposed the Neural Diving approach, which fixes most decision variables based on the initial solution prediction results obtained through a GNN. Consequently, a large-scale MILP is transformed into a smaller-scale MILP consisting of the remaining subset of decision

---

[*]Corresponding Author: Hua Xu (xuhua@tsinghua.edu.cn)

variables, effectively reducing the dimensionality of decision variables in the MILP. Sonnerat et al. (2021) further introduced NeuralLNS, which trains a Neural Selection policy to select the search neighborhood in Large Neighborhood Search (LNS) algorithm (Song et al., 2020; Ye et al., 2023a) to improve the initial solution obtained from Neural Diving. However, Neural Diving cannot fully exploit the embedding spatial information, and NeuralLNS heavily relies on large-scale solvers, leading to performance bottlenecks and limitations in solving capabilities constrained by the current solvers. In light of these limitations, Ye et al. (2023b) proposed a GNN&GBDT-guided optimizing framework that respectively employs the Multitask GNN to generate the embedding space, the Gradient Boosting Decision Tree (GBDT) to effectively use the embedding spatial information, and the Neighborhood Optimization to improve the current solution by means of a small-scale optimizer.

While the GNN&GBDT-guided framework has shown promising performance in practical applications, it still exhibits several noteworthy limitations. Firstly, representing MILPs as an entire graph poses challenges regarding model training and computational resources, particularly when tackling large-scale MILPs. Secondly, the GNN requires large-scale MILP instances of a similar size as training data, leading to significant computational and storage resource demands during the training phase. Thirdly, the application of problem reduction solely focuses on the decision variable level, overlooking potential synergies with constraint reduction, resulting in limited effectiveness in problem reduction. Generally, the performance of this method severely decreases when dealing with large-scale MILPs with complex constraints.

To address the challenges above, we propose Light-MILPopt, a lightweight optimization framework explicitly designed for large-scale MILPs, which consists of four stages: 1) *Problem Formulation.* Initially, we represent the MILP as a bipartite graph and then employ the FENNEL graph partition algorithm to divide the problem to reduce computational costs. 2) *Model-based Initial Solution Prediction.* We leverage the Edge Aggregated Graph Attention Network with half-convolutions, trained by a small-scale dataset with homogeneous structures, to predict and construct the initial solution for each divided subproblem. 3) *Problem Reduction.* We selectively reduce decision variables based on the generalized confidence threshold while simultaneously reducing constraints using the K-Nearest Neighbor (KNN) strategy. 4) *Data-driven Optimization.* Building upon problem division and reduction, we employ subgraph clustering and active constraint updating to guide neighborhood search and individual crossover, iteratively enhancing the current solution with a lightweight optimizer.

To validate the effectiveness and efficiency of Light-MILPopt which only uses a lightweight optimizer and a small-scale training dataset, we conduct extensive experiments on four large-scale benchmark MILPs and a real-world case study. The results indicate that, compared with the state-of-the-art optimization frameworks and large-scale solvers (e.g., Gurobi, SCIP), the proposed framework exhibits significant advantages when solving large-scale MILP. Further analyses show that the proposed framework can effectively reduce the model's computational complexity and achieve substantial problem reduction, which verifies its effectiveness and efficiency.

Our contributions can be summarized as follows.

- We propose the first lightweight framework that solves large-scale MILPs with only small-scale training datasets and lightweight optimizers, introducing Problem Formulation, Model-based Initial Solution Prediction, Problem Reduction, and Data-driven Optimization to reduce the computational complexity and improve the problem reduction capability.
- We demonstrate the effectiveness of the proposed framework in solving large-scale MILPs with lightweight optimizers and small-scale training datasets through a comparative analysis with state-of-the-art optimization frameworks and advanced solvers, providing initial insights into efficiently solving large-scale MILPs with limited computational resources.

## 2 PRELIMINARIES

### 2.1 MIXED INTEGER LINEAR PROGRAMS

Mixed Integer Linear Programs (MILPs) are a type of problem in which the objective function is linear under several linear constraints. In these problems, some or all decision variables are restricted to take integer values. Formally, an MILP has the form as the following (Achterberg, 2007).

$$\min_x c^T x, \text{subject to } Ax \leq b, l \leq x \leq u, x_i \in \mathbb{Z}, i \in \mathbb{I}, \tag{1}$$

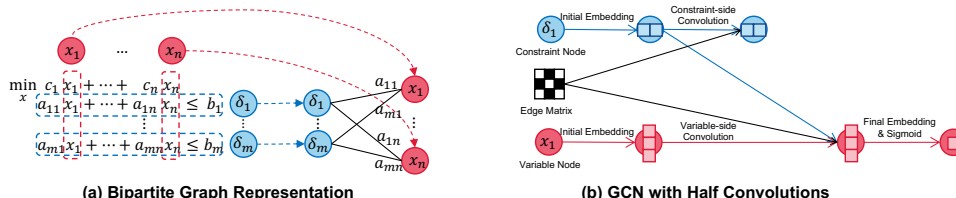

**(a) Bipartite Graph Representation**    **(b) GCN with Half Convolutions**

Figure 1: (a) Transforming an MILP instance to a bipartite graph. The set of $m$ constraint nodes $\{\delta_1, \ldots, \delta_m\}$ and the set of $n$ decision variables nodes $\{x_1, \ldots, x_n\}$ form the left-side constraint nodes set and the right-side variable nodes set of the bipartite graph representation. (b) The architecture of GCN with half-convolutions only has one layer. The layers in GCN can be broken down into two successive passes, one from variables to constraints and one from constraints to variables.

where $x$ denotes the decision variables whose number is denoted by $n \in \mathbb{Z}$, with $l, u, c \in \mathbb{R}^n$ being their lower bound, upper bound and coefficient, respectively. $A \in \mathbb{R}^{m \times n}$ and $b \in \mathbb{R}^m$ denote the linear constraints. $\mathbb{I} \subseteq \{1, 2, \ldots, n\}$ is the index set of integer variables. A solution is feasible for the MILP if decision variables $x \in \mathbb{R}^n$ satisfy all the constraints in Equation (1). A feasible solution is optimal if it attains the minimum objective function value of the minimized MILP (Schrijver, 1998). A constraint is an *active constraint* if the optimal solution of the MILP changes when removing this constraint, and vice versa for redundancy constraint(Bailey & Gillett, 1980; Murty & Yu, 1988).

## 2.2 BIPARTITE GRAPH REPRESENTATION

The bipartite graph representation for MILPs proposed by Gasse et al. (2019) achieves a lossless graphical representation of an MILP as the input of the neural embedding network (Nair et al., 2020), described in the left side of Figure 1. The $n$ decision variables in the MILP can be represented as the right-side variable nodes set in the bipartite graph, while the $m$ linear constraints can be represented as the left-side constraint nodes set. The edge connecting a variable node and a constraint node represents the corresponding variable that appears in that constraint. The details of the feature selection policy are shown in Appendix A.1.

## 2.3 GRAPH CONVOLUTIONAL NETWORK

In MILPs, based on the bipartite graph representation, the Graph Convolutional Network (GCN) (Kipf & Welling, 2016) is used for learning neural embedding and model-based initial solution prediction. Formally, let $\mathcal{E}$ represent the edges in a bipartite graph, and a $k$-layer GCN is as follows.

$$h_v^k = f_2^k(\{h_v^{(k-1)}, f_1^k(\{h_u^{(k-1)} : (u,v) \in \mathcal{E}\})\}), \tag{2}$$

where $h_v^k$ represents the hidden state of node $v$ in the $k$-th layer. The function $f_1^k$ combines the hidden values of the neighbors from the previous $(k-1)$-th layer to obtain aggregation information, while the function $f_2^k$ combines the hidden value of the current node $v$ with the aggregation information from its neighbors. For the bipartite graph representation, the GCN with two interleaved half-convolution layers can achieve better performance (Gasse et al., 2019; Yoon, 2022). Formally, let $\mathcal{V}_x$ denote the set of $n$ variable nodes and $\mathcal{V}_\delta$ denote the set of $m$ constraint nodes, and a $k$ layer half-convolutions GCN could be written as the following.

$$h_{\delta_j}^k = f_\delta^k(\{h_{\delta_j}^{(k-1)}, \sum_{(x_i, \delta_j) \in \mathcal{E}} g_\delta^k(\{h_x^{(k-1)}, h_{\delta_j}^{(k-1)}\})\}), \delta_j \in \mathcal{V}_\delta,$$

$$h_{x_i}^k = f_x^k(\{h_{x_i}^{(k-1)}, \sum_{(x_i, \delta_j) \in \mathcal{E}} g_x^k(\{h_{x_i}^{(k-1)}, h_{\delta_j}^k\})\}), x_i \in \mathcal{V}_x, \tag{3}$$

where $h_x^k, h_\delta^k, g_x^k$ and $g_\delta^k$ are aggregation functions. The right side of Figure 1 provides an overview of the 1-layer GCN with half-convolutions layers.

## 2.4 EDGE AGGREGATED GRAPH ATTENTION NETWORK

The current GCN model mentioned in Sec. 2.3 faces two significant problems. Firstly, it only uses a fixed policy to aggregate node information. Secondly, it can not fully incorporate the edge

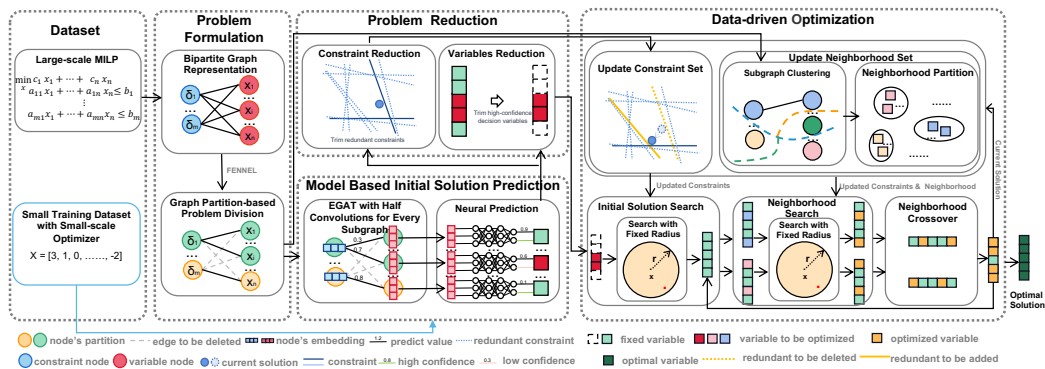

Figure 2: An overview of *Light-MILPopt*, where the blue line indicates that the component is only used during training. Firstly, the MILP is represented as a bipartite graph, and the FENNEL graph partitioning algorithm is employed for problem division to reduce model computational costs. Secondly, given the divided graph representations, the EGAT, trained by a small-scale training dataset, is used for predicting and constructing the initial solution of the original large-scale MILP. Thirdly, leveraging the predicted solution, the decision variable's confidence threshold and constraint's KNN strategy are introduced for problem reduction. Finally, based on problem division and reduction, the subgraph clustering and active constraint updating guide neighborhood search and individual crossover, iteratively improving the current solution by means of a lightweight optimizer.

features, ultimately resulting in poor performance in graphs with edge weights. To tackle these problems, Gong & Cheng (2019) proposed the Edge Aggregated Graph Attention Network (EGAT), which introduces an attention mechanism on graph neighborhoods and adequately exploits edge information, benefiting the learning neural embedding and model-based initial solution prediction. Formally, let $\mathcal{E}$ represent the edges in the bipartite graph. A $k$-layer EGAT could be written as below.

$$h_v^k = \sigma(\{\alpha_{uv}^k(h_u^{k-1}, \mathcal{E}_{uv}^{k-1}) : (u,v) \in \mathcal{E}\}, g^k(h_v^{k-1})), \tag{4}$$

where $h_v^k$ is the hidden state of node $v$ in the $k$-th layer; $\mathcal{E}_{uv}^k$ denotes the hidden state of the edge connecting $u$ and $v$ in the $k$-th layer; $\sigma$ is a non-linear activation; $g^k$ is a transformation that maps the node features from the input space to the output space. The $g^k$ is defined as: $g^k(h_v^{k-1}) = Wh_v^{k-1}$, where $W$ is a transformation parameter matrix. Additionally, $\alpha_{uv}^k$ is the attention coefficient that is a function of $h_u^{k-1}$, $h_v^{k-1}$ and $\mathcal{E}_{uv}^{k-1}$, and is defined as follows.

$$\alpha_{uv}^k = \text{DS}(exp\{L(a^T[Wh_u^{k-1}||a^T[Wh_v^{k-1}])\}\mathcal{E}_{uv}^{k-1}), \tag{5}$$

where DS is the doubly stochastic normalization operator defined in Appendix A.3; $L$ is the LeakyReLU activation function (Xu et al., 2020); $W$ is a transformation parameter matrix; and $||$ is the concatenation operation. Finally, the attention coefficients are used as new edge features for the next layer, defined as: $\mathcal{E}_{uv}^k = \alpha_{uv}^k$.

## 3   THE PROPOSED LIGHT-MILPOPT

This section describes *Light-MILPopt*, the proposed lightweight optimization framework that solves large-scale mixed integer linear programs with lightweight optimizer and small-scale training dataset. *Light-MILPopt* can be divided into four stages: *Problem Formulation* (Sec. 3.1), *Model-based Initial Solution Prediction* (Sec. 3.2), *Problem Reduction* (Sec. 3.3), and *Data-driven Optimization* (Sec. 3.4). We present the overall architecture of the proposed framework in Figure 2.

### 3.1   PROBLEM FORMULATION

In light-MILPopt, the large-scale MILP is represented in the form of a *Bipartite Graph* at first. Then, the FENNEL *Graph Partition* algorithm (Tsourakakis et al., 2014) is applied to divide the graph into several blocks. Based on the above steps, all the subgraphs obtained from the graph partition form the inputs for feature-embedding neural networks.

**Bipartite Graph Representation**. Based on the classic bipartite graph representation introduced in Sec. 2.2, the feature selection policy is further improved to enhance embedding ability. For one thing, the random feat strategy (Chen et al., 2023) is introduced to get better representation and embedding capabilities when facing some particular MILPs called "foldable" (Chen et al., 2023). Specifically, by adding random features, the predicting of MILP feasibility, optimal objective values, and optimal solutions becomes more reliable. For another thing, the classic feature selection policy for categorical features, which only uses a single integer is not sensitive to categorical information, resulting in weak feature neural embedding in feature-embedding neural networks. Therefore, the one-hot strategy (Shen et al., 2022) is utilized to represent the categorical features, further enhancing the impact of categorical information on the neural encoding results. More details of the introduced new policy are shown in the Appendix A.2.

**Graph Partition-based Problem Division**. For an MILP, the size of the bipartite graph representation is linearly correlated with the problem scale (Gasse et al., 2019). Therefore, when dealing with large-scale MILPs with millions of variables, directly using the entire bipartite graph representation as the input for feature-embedding neural networks for training or predicting would lead to significant demands on computational and storage resources. Inspired by the Graph-Bert (Zhang et al., 2020) subgraph partitioning concept, we use the FENNEL graph partitioning algorithm, described in Appendix A.4, to divide the original entire bipartite graph into several low-correlation subgraphs, with each subgraph representing a divided subproblem. The partitioned subgraphs are then sequentially fed into the feature-embedded neural network, thus transforming the solution of a large-scale optimization problem into the parallel solution of multiple small-scale problems.

## 3.2 MODEL-BASED INITIAL SOLUTION PREDICTION

Given the graph representation with multiple small-scale subgraphs for the large-scale MILP, *EGAT with Half-convolutions* learns the neural embedding for the decision variables. Then the *Neural Prediction* network with Multi-Layer Perceptron (MLP) structure predicts the initial value of the corresponding decision variable in the MILP through the neural embedding. Finally, the predicted initial solution will guide the subsequent problem reduction in the next stage.

**EGAT with Half-convolutions**. Based on the GNN with half-convolutions (Yoon, 2022) (Sec. 2.3) and EGAT (Gong & Cheng, 2019) (Sec. 2.4), we propose an EGAT with multi-layer half-convolutions structure which combines the advantages of the above two methods, intended to further improve the neural embedding learning of the decision variables for every divided subgraph. Formally, based on Equation (3) and (4), letting $\mathcal{E}$ represent the edges in the bipartite graph, a $k$-layer EGAT with multi-layer half-convolutions structure can be represented as follows.

$$
\begin{aligned}
\alpha_{x_i \delta_j}^k &= \mathrm{DS}(exp\{L(a^T[Wh_{x_i}^{k-1}||a^T[Wh_{\delta_j}^{k-1}])\}\mathcal{E}_{x_i \delta_j}^{k-1}), \\
h_{\delta_j}^k &= \sigma(\{\alpha_{x_i \delta_j}^k(h_x^{(k-1)}, \mathcal{E}_{x_i \delta_j}^{k-1}) : (x_i, \delta_j) \in \mathcal{E}\}, g^k(h_{\delta_j}^{k-1})), \\
h_{x_i}^k &= \sigma(\{\alpha_{x_i \delta_j}^k(h_{\delta_j}^k, \mathcal{E}_{x_i \delta_j}^{k-1}) : (x_i, \delta_j) \in \mathcal{E}\}, g^k(h_{x_i}^{k-1})), \\
\mathcal{E}_{x_i \delta_j}^k &= \alpha_{x_i \delta_j}^k,
\end{aligned}
\tag{6}
$$

where $h_{\delta_j}^k$ and $h_{x_i}^k$ represent the hidden state of constraint node $\delta_j$ and variable node $x_i$ in the $k$-th layer, respectively; $\mathcal{E}_{x_i \delta_j}^k$ and $\alpha_{x_i \delta_j}^k$ denotes the hidden state and attention coefficient of the edge connecting variable node $x_i$ and constraint node $\delta_j$ in the $k$-th layer, respectively; $\sigma$ is a non-linear activation; $g^k$ is a transformation that maps the node features from the input space to the output space. The information transition flow of EGAT with half-convolutions is shown in Figure 3.

**Neural Prediction**. Based on the neural embedding of the decision variables obtained by the EGAT, a $p$-layer MLP is used to predict the initial value of the corresponding decision variable for every divided small-scale MILPs. For binary variables $x_i$, the neural embedding of the decision variable only undergoes a single Multi-Layer Perceptron (MLP) transformation, followed by the application of the Sigmoid activation function to the MLP's output. This process allows the output to convey the probability that the respective decision variable equals 1, denoted as $p(x_i = 1|M)$. For general integer variables or even real variables, according to the required accuracy, the neural embedding of a decision variable needs to pass through multiple independent MLPs with Sigmoid activation function in parallel. For the $i$-th decision variable, the output of the $j$-th MLP can represent the probability that the $j$-th binary bit is 1, represented as $p(x_{ij} = 1|M)$, and the value of the bit is taken

Figure 3: Information transition flow in the EGAT with half-convolutions. The information transitions run consecutively as follows: *Step 1*, transforming variable nodes and constraint nodes information to the edge; *Step 2*, transforming the variable nodes and edge information to constraint nodes; *Step 3*, transforming constraint nodes and edge information to the variable node.

$\lfloor p(x_{ij} = 1|M) + 0.5 \rfloor$, where Focal Loss (Lin et al., 2017) is used for training. Due to the weak correlation among the small-scale MILPs obtained by problem division, all the split small-scale MILP have obtained initial predicted solutions, the initial predicted solutions of the split small-scale MILP can be concatenated to obtain the initial predicted solutions of the original large-scale MILP.

## 3.3 PROBLEM REDUCTION

Given the predicted initial solution of the MILP, the generalized confidence threshold method adaptively fixes the high-confidence decision variable to achieve *Variables Reduction*. Then, KNN strategy is used for *Constraint Reduction* to identify active constraints. The unfixed decision variables and KNN constraints can jointly guide the initial solution search and iterative optimization.

**Variables Reduction.** Classic approaches employ SelectiveNet (Geifman & El-Yaniv, 2019) to predict the initial solution of MILP while simultaneously predicting the variable reduction based on the target reduction ratio. However, this method requires multiple training runs for different target reduction ratios, leading to additional training overhead. Therefore, Light-MILPopt proposes a generalized confidence threshold method based on the confidence threshold method specifically designed for binary variables (Yoon, 2022), aiming to reduce efficient variable dimensionality. Specifically, based on the MILP's initial solution obtained from the model predictions, the confidence value $f_i$ of the decision variable $x_i$ which contains $c$ binary bits can be represented as follows.

$$f_i = \sqrt[c]{\prod_{j=1}^{c} \max[p(x_{ij} = 1|M), 1 - p(x_{ij} = 1|M)]}. \tag{7}$$

Next, the confidence values of the decision variables are sorted in descending order. Based on the desired reduction ratio of $k\%$, the top $k\%$ of the decision variables are fixed to their predicted values. The remaining decision variables form a new small-scale optimization problem that is fed into the initial solution search of the next stage, achieving variables reduction for the large-scale MILP.

**Constraint Reduction.** The constraints in MILP can be classified into redundant constraints and active constraints as mentioned in Sec. 2.1. Effective removal of redundant constraints simplifies the solution space and speeds up the process of solving large-scale MILPs, termed as constraints reduction. When without considering the integer constraints, the feasible domain formed by linear constraints in MILP possesses the property of convexity. If the optimal solution of the MILP is known, active constraints are often those that are close to the optimal solution (Runarsson & Yao, 2005). Since the predicted initial solution is the result of fitting a model to the optimal solution, the predicted initial solution can be treated as an estimation of the optimal solution, utilized for computing the distance from each constraint hyperplane. Finally, according to the calculation results, the KNN constraints of the predicted initial solution are selected as the prediction of the active constraints, and the remaining redundant constraints are deleted to realize constraint reduction.

## 3.4 DATA-DRIVEN OPTIMIZATION

Based on the predicted initial solution and the problem reduction, we first solve the reduced subproblem to obtain the *Initial Solution* for the complete MILP. Then, under the guidance of *Neighborhood Set Updating* and the active *Constraint Set Updating*, neighborhood search and individual crossover

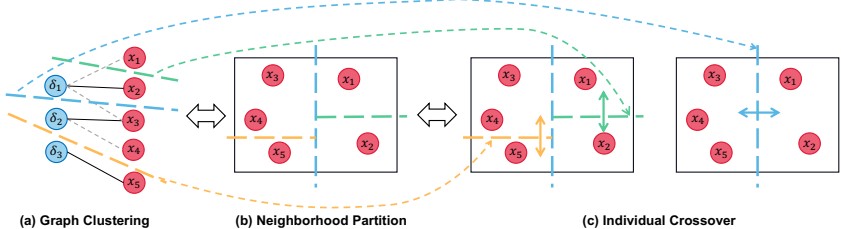

**(a) Graph Clustering**          **(b) Neighborhood Partition**          **(c) Individual Crossover**

Figure 4: (a) Apply random-cluster algorithm to the divided subgraphs to group subgraphs into several clusters. (b) Decision variables in a cluster form a neighborhood block. (c) Employ the hierarchical crossover strategy for neighborhood merging and corresponding individual crossover.

iteratively improve the current solution. Finally, when a predetermined wall-clock time or condition is reached, the current solution is output as the final optimization result.

**Initial Solution Search.** For the large-scale MILP with $n$ decision variables, we first define a coefficient $\alpha \in (0, 1)$ to denote that the lightweight optimizer can solve small-scale MILP containing at most $\alpha n$ decision variables. Following this, an initial solution search method described in Appendix A.6 is employed for the reduced small-scale MILP obtained from the stage of dimensionality reduction. Finally, the optimal solution obtained by the lightweight optimizer is recombined with the reduced decision variables to form an initial feasible solution.

**Neighborhood Set Update.** In each iteration of optimization, both the neighborhood search and individual crossover require a new neighborhood to prevent getting stuck in local optima. Based on the subgraph partitioning obtained from the FENNEL graph partitioning algorithm, we use random-cluster algorithm (Grimmett, 2006) to further group the subgraphs into several clusters that are suitable for the small-scale solver, shown in Figure 4(a) and Appendix A.7. Each cluster contains decision variables from multiple subgraphs, forming a neighborhood block. The graph cluster corresponding to neighborhood set is shown schematically in Figure 4(b).

**Constraint Set Update.** Due to the potential distance between the initial predicted solution and the optimal solution, the initial prediction of active constraints is biased. Therefore, after each optimization iteration, the current solution gradually approaches the optimal solution, and Light-MILPopt calculates a new set of KNN constraints to obtain more accurate estimates of active constraints using the new current solution as the reference point. Furthermore, since the prediction of active constraints becomes more and more reliable as the optimization proceeds, we adopt a progressive strategy by choosing a larger value of $K$ in KNN at the beginning and gradually narrowing down the set of active constraints. Specifically, after each iteration, $K$ is updated to $K * \eta$ until the predefined minimum reduction threshold is reached, where $\eta \in (0, 1)$ is the preset descent rate, contributing to the progressive reduction of constraints. Finally, as the active constraint set changes, the feasible solution region is also altered. So we employ the REPAIR algorithm described in Appendix A.8 to "repair" the current solution, which ensures the current solution falls within the new feasible solution range changes by the new active constraint set.

**Iterative Optimization.** In each iteration, based on the updated neighborhood set and updated active constraint set, the neighborhood search is parallelly executed in each neighborhood, utilizing a lightweight optimizer. Then the results of the neighborhood search are used as individuals for hierarchical crossover shown in Figure 4(c), allowing the integration and inheritance of strengths from different neighborhoods, leading to improved solutions efficiently. When the current iteration ends, the results of the individual crossover update the current solution and simultaneously update the neighborhood set and the active constraint set. Finally, when meeting the predetermined time or conditions, the current solution is output as the final optimization result.

## 4 EXPERIMENTS

To validate the effectiveness and efficiency of the proposed Light-MILPopt for large-scale MILPs, we compare it with two types of baselines on four widely used large-scale NP-hard standard benchmark MILP datasets: Set Covering (SC, Minimize) (Caprara et al., 2000), Minimum Vertex Cover

Table 1: Comparison of objective value results with baselines under the same running time on different large-scale MILPs. Ours-30%S means the proposed framework with the scale-limited versions of SCIP which limit the variable proportion $\alpha$ to 30%. GBDT-50%G means the GNN&GBDT framework with the scale-limited versions of Gurobi which limit the variable proportion $\alpha$ to 50%. ↑ means the result is better than the baseline. - means that no feasible solution is found.

| | $SC_1$ | $SC_2$ | $MVC_1$ | $MVC_2$ | $MIS_1$ | $MIS_2$ | $MIKS_1$ | $MIKS_2$ | Case Study |
|---|---|---|---|---|---|---|---|---|---|
| Ours-30%S | 17121.5↑ | 166756.0↑ | 27337.8↑ | 273014.6↑ | 22621.7↑ | 227074.5↑ | 35067.8↑ | 355887.6↑ | 944086.4↑ |
| Ours-30%G | **17047.3↑** | **163975.9↑** | **27223.3↑** | **272579.5↑** | **22658.0↑** | **227305.4↑** | **35533.4↑** | **357439.5↑** | **979797.8↑** |
| GBDT-30%S | 17222.2 | 261174.0 | 27515.4 | 276306.9 | 22389.3 | 223349.8 | - | - | - |
| GBDT30%G | 18487.6 | 281021.2 | 27700.8 | 281234.5 | 22115.9 | 210019.2 | - | - | - |
| Ours-50%S | 16147.2↑ | 166966.9↑ | 26956.8↑ | 269771.3↑ | 22963.6↑ | 230278.1↑ | **36125.5↑** | 357483.8↑ | 944166.1↑ |
| Ours-50%G | **16108.1↑** | **160015.5↑** | **26950.7↑** | **269571.5↑** | **22966.5↑** | **230432.7↑** | 36108.2↑ | **362265.1↑** | **980688.0↑** |
| GBDT50%S | 16728.8 | 268294.9 | 27107.9 | 271777.2 | 22795.7 | 227006.4 | - | - | - |
| GBDT50%G | 17503.4 | 252797.2 | 27329.9 | 274600.8 | 22530.1 | 215393.6 | - | - | - |
| SCIP | 25191.2 | 385708.4 | 31275.4 | 491042.9 | 18649.6 | 9104.3 | 29974.7 | 168289.9 | 924954.5 |
| Gurobi | 17934.5 | 320240.4 | 28151.3 | 283555.8 | 21789.0 | 216591.3 | 32960.0 | 329642.4 | - |
| Time | 2000s | 12000s | 2000s | 8000s | 2000s | 8000s | 2000s | 6000s | 1000s |

(MVC, Minimize) (Dinur & Safra, 2005), Maximum Independent Set (MIS, Maximize) (Tarjan & Trojanowski, 1977), Mixed Integer Knapsack Set (MIKS, Maximize) (Atamtürk, 2003) and one real-world large-scale MILP in the internet domain (Case Study, Maximize). One type of the baselines is the state-of-the-art MILP solvers, including SCIP (Achterberg, 2009) and Gurobi (Achterberg, 2019). The other one is the latest state-of-the-art ML-based optimization framework based on GNN&GBDT(Ye et al., 2023b). In Appendix B, we present more details of the datasets, comprehensive experimental setup, and the baselines.

To make fair comparisons, we use multiple evaluation metrics to study the performances of all the related methods in this paper, including comparisons of solving effectiveness under the same running time (Sec. 4.1), comparisons of solving efficiency under the same solving results (Sec. 4.2), and analysis of convergence (Sec. 4.3). Additional experimental results are detailed in Appendix C. Code for reproducing all the experiments can be found at https://github.com/thuiar/Light-MILPopt.

### 4.1 COMPARISONS OF SOLVING EFFECTIVENESS

To verify the effectiveness of the proposed Light-MILPopt, we compare the solving results of the proposed framework with the large-scale solvers SCIP, Gurobi, and the GNN&GBDT frameworks, at the same runtime. Light-MILPopt only uses a lightweight optimizer which limits the variable proportion $\alpha$ to 30% and 50%, and using only 1% of the size of large-scale benchmark MILPs for training data. We present the experimental results in Table 1. On the one hand, compared to the large-scale baseline solvers SCIP and Gurobi, Light-MILPopt obviously outperforms them only using a scale-limited version solver with variable proportion $\alpha = 30\%$. On the other hand, the proposed framework achieves better results than the GNN&GBDT frameworks in integer programs with the same scale of variable reduction, efficiently solving large-scale MILPs, which cannot be solved by the GNN&GBDT framework.

It is worth noting that in the SC scenario, Light-MILPopt outperforms all baselines significantly. Upon in-depth analysis, it becomes apparent that the small-scale subgraphs obtained through problem division contribute to the improved performance and efficiency of initial solution prediction. Additionally, SC is a problem characterized by intricate constraints, and constraint reduction decreases the time required for each iteration of optimization to $1/5$ of the original time. This factor greatly contributes to the remarkable solving ability of Light-MILPopt for this particular problem.

### 4.2 COMPARISONS OF SOLVING EFFICIENCY

To further validate the efficiency of the Light-MILPopt, we compare the running time of the proposed framework with the baseline algorithms with fixed-solving results. We represent the experimental results in Table 2. It's evident that Light-MILPopt significantly reduces the time required to obtain the same optimization results compared to all the baseline approaches on all MILPs. Specifically, compared to the large-scale baseline solvers, the proposed framework can achieve the same results in only $0.5\%$ of the time for the benchmark MILPs, including $SC_1$, $MVC_1$, $MIS_1$ and $MIKS_1$. Even compared to the state-of-the-art ML-based frameworks, our Light-MILPopt can save more than $90\%$ of the solution time on most MILPs to achieve the same results. It is interesting to note that for $MVC_1$ and $MIS_1$, while there is little difference among all methods in terms of achieving the same

Table 2: Comparison of running time with SCIP, Gurobi and GNN&GBDT framework under the same target value on different large-scale MILPs. Ours-30%S means the proposed framework with the scale-limited versions of SCIP which limit the variable proportion $\alpha$ to 30%. GBDT-50%G means the GNN&GBDT framework with the scale-limited versions of Gurobi which limit the variable proportion $\alpha$ to 50%. ↑ means the result is better than the baseline. $>$ means indicates the inability to achieve the target value in some instances within the maximum running time.

| | $\mathbf{SC}_1$ | $\mathbf{SC}_2$ | $\mathbf{MVC}_1$ | $\mathbf{MVC}_2$ | $\mathbf{MIS}_1$ | $\mathbf{MIS}_2$ | $\mathbf{MIKS}_1$ | $\mathbf{MIKS}_2$ | **Case Study** |
|---|---|---|---|---|---|---|---|---|---|
| Ours-30%S | 1998.1s↑ | 11823.0s↑ | 1951.6s↑ | 7967.2s↑ | 1951.6s↑ | 7967.2s↑ | 1982.0s↑ | 11980.4s↑ | 996.4s↑ |
| Ours-30%G | **1166.8s↑** | **5645.0s↑** | **1475.3s↑** | **6453.3s↑** | **1487.3s↑** | **7250.5s↑** | **593.9s↑** | **7941.9s↑** | **511.5s↑** |
| GBDT-30%S | >48369.2s | >60000s | >60000s | >60000s | >60000s | >60000s | - | - | - |
| GBDT30%G | >30347.8s | >60000s | >60000s | >60000s | >60000s | >60000s | | | |
| Ours-50%S | 352.2s↑ | 11441.3s↑ | 203.1s↑ | 1815.3s↑ | 225.9s↑ | **1945.7s↑** | 194.9s↑ | 9576.1s↑ | 776.2s↑ |
| Ours-50%G | **177.8s↑** | **1795.4s↑** | **193.8s↑** | **1503.3s↑** | **223.5s↑** | 2062.7s↑ | **160.5s↑** | **2137.8s↑** | **506.9s↑** |
| GBDT50%S | 587.6s | >60000s | 297.6s | 7570.5s | 348.6s | 5920.7s | - | - | - |
| GBDT50%G | 5041.6s | >60000s | 29320.5s | 21397.3s | 4227.1s | 27952.9s | - | - | - |
| SCIP | >60000s | >60000s | >60000s | >60000s | >60000s | >60000s | >60000s | >60000s | 3097.0s |
| Gurobi | >60000s | >60000s | >60000s | >60000s | >60000s | >60000s | 45599.4s | >60000s | 2584.7s |
| Target | 17121.5 | 166756.0 | 27337.8 | 273014.6 | 22621.7 | 227074.5 | 35067.8 | 355887.6 | 944086.4 |

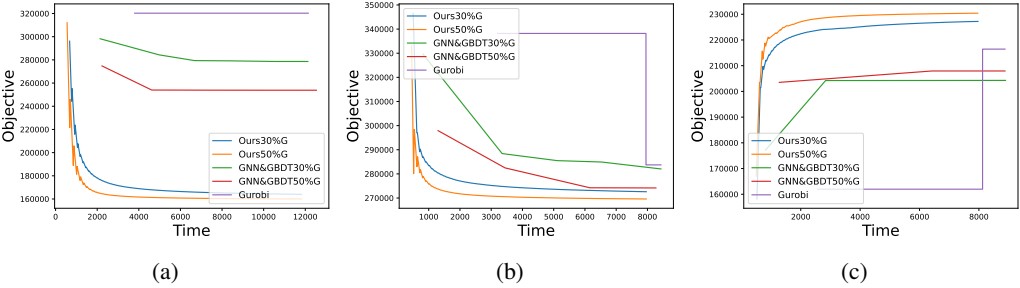

(a)       (b)       (c)

Figure 5: Time-objective figure for three of the benchmark MLIPs. (a) The minimized SC problem. (b) The minimized MVC problem. (c) The maximized MIS problem.

optimization results within a certain time frame, there is a significant difference in running time. Our analyses demonstrate that the proposed framework can significantly reduce the time required to achieve the same results as the baseline methods.

## 4.3 ANALYSIS OF CONVERGENCE

Convergence is an essential metric for evaluating the performance of optimization frameworks. To analyze the convergence of Light-MILPopt, we record the trend of the objective value with the iteration time of the proposed framework and baseline algorithm on three MILPs that the GNN&GBDT framework can solve, including SC, MVC and MIS. We visualize the time-objective variation in Figure 5. We can see that the proposed framework can obtain high-quality solutions for large-scale MILPs with only small-scale training data and a lightweight optimizer. Figure 5 also demonstrates that the convergence performance of Light-MILPopt is not weaker than that of the state-of-the-art solver Gurobi as well as the state-of-the-art ML-based optimization framework.

## 5 CONCLUSION

This paper proposes Light-MILPopt, a lightweight optimization framework for large-scale MILPs. Light-MILPopt uses graph partition-based problem division and EGAT with half-convolutions to efficiently predict initial MILP solutions with only a lightweight training dataset. Through variables and constraints reduction, Light-MILPopt rapidly updates the current solution with a lightweight optimizer. Experimental evaluations conducted on four standard large-scale MILPs and a real-world case study demonstrate that our framework outperforms SCIP, Gurobi, and the GNN&GBDT-based optimization framework. In the future, we will further improve the proposed framework and explore its applicability in ultra-large-scale, multi-objective, and nonlinear constraint scenarios.

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

**APPENDIX**

This Appendix contains three sections. Appendix A introduces the details of some algorithms mentioned in the main text, elaborating on their implementation details through formulas or pseudo-code. Appendix B describes the experimental setup, including the standard mathematical form of the standard benchmark MILP, parameter settings in data generation, and other relevant setup information. Appendix C shows supplementary experimental results to verify the effectiveness and efficiency of Light-MILPopt further. Appendix D shows more additional information on mixed-integer linear programms.

## A  ADDITIONAL ALGORITHM DETAILS

### A.1  FEATURE SELECTION POLICY

In the classical feature selection policy (Gasse et al., 2019; Nair et al., 2020), the feature selection of nodes and edges usually depends on the coefficients in the formulation of MILP. Formally, let $h_x^i, h_\delta^i, h_{(i,j)}$ denote the feature selection of the $i$-th variable node, $j$-th constraint node and edge $(i,j)$, the classical feature selection policy can be written as the following.

$$
\begin{aligned}
h_x^i &= (c_i, l_i, u_i, t_i, d_i), \\
h_\delta^j &= (b_j, \mathfrak{o}_j), \\
h_{(i,j)} &= (a_{ij}),
\end{aligned}
\tag{8}
$$

where $c_i, k_i, u_i, t_i$ denotes the coefficient, lower bound and upper bound of the $i$-th decision variable respectively; $d_i \in \{0, 1\}$ represents whether the $i$-th decision variable restricted to take integer value or not; $b_j, \mathfrak{o}_j$ refers the value and symbol of the $j$-th constraint; $a_{ij}$ denotes the weight of edge $(i, j)$.

### A.2  PROPOSED FEATURE SELECTION POLICY

Based on the classic bipartite graph representation introduced in Appendix A.1, the feature selection policy is further improved to enhance embedding ability.

On one hand, the random feat strategy (Chen et al., 2023) is introduced to achieve better representation and embedding capabilities when facing some particular MILPs called "foldable" (Chen et al., 2023), that can rewrite Equation (8) as the following.

$$
\begin{aligned}
h_x^i &= (c_i, l_i, u_i, d_i, t_i, \xi), \\
h_\delta^j &= (b_j, \mathfrak{o}_j, \xi), \\
h_{(i,j)} &= (a_{ij}),
\end{aligned}
\tag{9}
$$

where $\xi \sim U(0, 1)$ is the random feat between 0 and 1. Specifically, by adding random features, the predicting of MILP feasibility, optimal objective values, and optimal solutions becomes more reliable. On the other hand, in the classical bipartite graph representation, for categorical features such as the type of decision variables and the type of constraints, a single integer is used to represent the categorical feature's type. For example, for the representation of the type of constraints in $j$-th constraint node, the feature selection policy is $\{\geq\to 0, \leq\to 1, =\to 2\}$. However, this feature selection policy is not sensitive to categorical information, resulting in weak feature neural embedding in feature-embedding neural networks. Therefore, the one-hot strategy that is widely used in the field of text feature representation is used for representing the categorical features, further enhancing the impact of categorical information on the neural encoding results. In this feature selection policy, for the representation of the type of constraints in $j$-th constraint node, the feature selection policy is changed to $\{\geq\to [1, 0, 0], \leq\to [0, 1, 0], =\to [0, 0, 1]\}$. So we can rewrite Equation (9) to obtain the final feature selection policy.

$$
\begin{aligned}
h_x^i &= (c_i, l_i, u_i, d_i, t_i, \xi), \\
h_\delta^j &= (b_j, \mathfrak{o}_j', \xi), \\
h_{(i,j)} &= (a_{ij}),
\end{aligned}
\tag{10}
$$

where $\mathfrak{o}_j'$ is the one-hot representation of the type of constraints in $j$-th constraint node, and we have $\mathfrak{o}_j' : \{\geq\to [1, 0, 0], \leq\to [0, 1, 0], =\to [0, 0, 1]\}$.

## A.3 DOUBLY STOCHASTIC NORMALIZATION OPERATOR

In graph convolution operations, we utilize edge feature matrices as filters to perform element-wise multiplication with the node feature matrix. To prevent an undesired increase in the magnitude of the output features due to this multiplication, it is essential to normalize the edge features. Let's denote the raw edge features as $\hat{E}$. We can obtain the normalized features, denoted as $E$, using the following procedure of doubly stochastic normalization operator.

$$\tilde{E}_{ijp} = \frac{\hat{E}_{ijp}}{\sum_{k=1}^{N} \hat{E}_{ikp}}, \tag{11}$$

$$E_{ijp} = \sum_{k=1}^{N} \frac{\tilde{E}_{ikp} \tilde{E}_{jkp}}{\sum_{v=1}^{N} \tilde{E}_{vkp}}, \tag{12}$$

where $E$ is a tensor representing the edge features of the graph; $E_{ij} \in \mathbb{R}^P$ represents the $P$-dimensional feature vector of the edge connecting the $i$-th and $j$-th nodes; $E_{ijp}$ denotes the $p$-th channel of the edge feature in $E_{ij}$. In addition, when the $i$-th and $j$-th points are not contiguous, $E_{ijp} = 0$ for any $p$. Such kind of normalized edge feature tensor $E$ has the following properties.

$$E_{ijp} \geq 0, \tag{13}$$

$$\sum_{i=1}^{N} E_{ijp} = \sum_{j=1}^{N} E_{ijp} = 1. \tag{14}$$

In simpler terms, the edge feature matrices $\hat{E}$, where $p$ ranges from 1 to $P$, are square, nonnegative real matrices in which both the rows and columns sum up to 1. This property classifies them as doubly stochastic matrices, meaning they exhibit both left and right stochastic characteristics. In mathematical terms, when you have a finite Markov chain with a transition matrix that is doubly stochastic, it will have a uniform stationary distribution. Now, in the context of a multi-layer graph neural network, these edge feature matrices undergo repeated multiplication across the layers. Utilizing doubly stochastic normalization can enhance stability in this process.

## A.4 FENNEL GRAPH PARTITION ALGORITHM

---
**Algorithm 1** FENNEL-based Graph Partition Algorithm
---
**Input:** The number of blocks $k$, graph $\mathcal{G} = (\mathcal{V}, \mathcal{E})$, parameters $\mu, \gamma$
**Init:** Label $\mathcal{F} = \{\}$, block $\mathcal{P} = \{\{\}, \ldots, \{\}\}$
$n \leftarrow |\mathcal{V}|$
$m \leftarrow |\mathcal{E}|$
$\alpha \leftarrow \sqrt{k} \frac{m}{n^{3/2}}$
load_limit $\leftarrow \nu \frac{n}{k}$
**for** $v = 1$ **to** $n$ **do**
  **for** $i = 1$ **to** $k$ **do**
    **if** $|P_i| <$ load_limit **then**
      $N(v) \leftarrow \{u, (u, v) \in \mathcal{E}\}$
      $\delta_g(v, \mathcal{P}_i) \leftarrow |\mathcal{P}_i \cap N(v)| - \alpha\gamma|P_i|^{\gamma-1}$
    **end if**
  **end for**
  ind $\leftarrow \arg\max_i \delta_g(v, \mathcal{P}_i)$
  Add $v$ into $\mathcal{P}_i$
  $\mathcal{F}[v] \leftarrow$ ind
**end for**
**Return:** $\mathcal{F}$
---

For graphs at the million-level scale, the streaming graph partitioning algorithm is commonly employed. There exist two primary heuristics for streaming graph partitioning. On one hand, newly arrived vertices are placed in the cluster with the greatest number of neighbors. On the other hand,

they can be placed in the cluster with the fewest non-neighbors. FENNEL, however, introduces an innovative approach by combining these two heuristics.

Formally, for a graph $\mathcal{G} = (\mathcal{V}, \mathcal{E})$ comprising $n$ vertices and $m$ edges, to be partitioned into $k$ blocks, FENNEL aggregates vertices with dense edges into a single block, while segregating vertices with sparse edges into separate blocks. More specifically, it systematically evaluates each newly arrived vertex $v$ in the graph, one by one, and computes $\delta_g(v, \mathcal{P}_i)$ for each block $\mathcal{P}_i$ that satisfies $|P_i| < \nu \frac{n}{k}$, as follows:

$$\delta_g(v, \mathcal{P}_i) \leftarrow |\mathcal{P}_i \cap N(v)| - \alpha\gamma|P_i|^{\gamma-1}, \tag{15}$$

where $N(v)$ denotes the neighbor node set of $v$, $\nu, \gamma$ are preset parameters related to block balancing and minimum cut, and $\alpha = \sqrt{k}\frac{m}{n^{3/2}}$ denotes the balance of two types of heuristics. The details are shown in Algorithm 1.

## A.5 INITIAL SOLUTION PREDICTION

The initial solution prediction based on machine learning often uses Graph Convolutional Neural Networks (GNN) to predict the initial solution of the complete problem (Gasse et al., 2019; Nair et al., 2020; Sonnerat et al., 2021; Ye et al., 2023b). However, this approach faces two major challenges. On one hand, as the problem scale increases, the required storage resources, especially GPU memory, steadily rise, making it difficult to solve large-scale or even super-large-scale problems. On the other hand, GNNs achieve initial solution prediction by learning the distribution mapping of isomorphic mixed-integer problems to optimal solutions. However, in more complex problems, the predicted solutions obtained using this method are often infeasible.

To overcome the shortcomings of existing methods, we propose a novel initial solution prediction strategy. To address the first problem, we propose using the FENNEL graph partitioning algorithm to decompose large-scale Mixed-Integer Linear Programming (MILP) into several smaller subproblems. We predict the optimal solution for each subproblem and concatenate them to obtain the initial feasible solution for the complete problem, allowing training and inference to be performed only on small-scale problems. Regarding the second problem, previous work attempted to predict the probability of each point being set to 1. Then, for parameters $(k_0, k_1)$, $k_0$ probabilities with the highest values are fixed to 1, and $k_1$ probabilities with the lowest values are fixed to 0 (Han et al., 2023). However, this method requires setting different hyperparameters for different problems. Therefore, for Mixed-Integer Linear Programming problems, we propose using the Repair algorithm to obtain a feasible solution from the current solution. This algorithm identifies illegally constrained decision variables, cancels the predictions for these variables, uses a small-scale solver to solve for the initial values of the canceled decision variables, and ultimately obtains a feasible solution. See Appendix Algorithm 5 for details on the Repair algorithm.

It's worth notice that the framework's solving approach is designed for problems with the same structure and mathematical properties. It involves unified training, prediction, and neighborhood search to obtain optimized solutions. Additionally, it can efficiently handle large-scale problems using only small-scale training data. For problems with different properties, a secondary training process is required to construct a new feasible solution prediction model tailored to the specific problem. For example, problems with different scales of minimum point cover can be trained together. However, if the goal shifts to solving minimum set cover problems, a new feasible solution prediction model needs to be retrained.

## A.6 INITIAL SOLUTION SEARCH

Given the predicted value $\hat{y}_i$ and the prediction loss $\mathcal{P}_i$ for each decision variable, the decision variables are arranged in ascending order based on their prediction losses. For the pre-defined coefficient $\alpha \in (0, 1)$ which denotes that the lightweight optimizer can solve small-scale MILP containing at most $\alpha n$ decision variables, the first $(1 - \alpha)n$ decision variables are held constant, while the remaining variables are explored within a predetermined fixed radius. The specific steps are outlined in Algorithm 2.

In Algorithm , $\eta \in (0, 1)$ is a reduction coefficient used to expand the fixed proportion, REPAIR () is the function shown in Appendix A.8.

---

**Algorithm 2** Initial Solution Search

---

**Input:** The number of decision variables $n$, predicted value $\hat{y}$, prediction loss $\mathcal{P}$, variable proportion $\alpha$
**Init:** Initial Solution $\mathcal{X} = \{\}$
$\mathcal{X} \leftarrow \hat{y}$
Sort the decision variables in ascending order of $\mathcal{P}$
$\alpha_{set} = \alpha$
**repeat**
    $\mathcal{F} \leftarrow$ The first $(1 - \alpha_{set})n$ decision variables  ▷Fixed
    $\mathcal{U} \leftarrow$ The last $\alpha_{set}$ decision variables         ▷Unfixed
    $\mathcal{F}', \mathcal{U}' \leftarrow \text{REPAIR}(\mathcal{F}, \mathcal{U}, \mathcal{X})$
    **if** $|\mathcal{U}'| > \alpha n$ **then**
        $\alpha_{set} = \eta * \alpha_{set}$
    **end if**
**until** $|\mathcal{U}'| \leq \alpha n$
$\mathcal{X} \leftarrow \text{SEARCH}(\mathcal{F}', \mathcal{U}', \mathcal{X})$
**Return:** $\mathcal{X}$

---

### A.7 NEIGHBORHOOD SEARCH AND INDIVIDUAL CROSSOVER

Specifically, for the $i$-th neighborhood $N_i$, Algorithm 3 shows the details in neighborhood search.

---

**Algorithm 3** Neighborhood Search

---

**Input:** The set of decision variables $X$, the number of decision variables $n$, predicted value $\hat{y}$, prediction loss $\mathcal{P}$, variable proportion $\alpha$, neighborhood $N_{now}$, current solution $\mathcal{X}$
**Init:** Neighborhood search solution $\mathcal{X}' = \{\}$
Sort the decision variables in $N_{now}$ in descending order of $\mathcal{P}_i * |\phi_i - \hat{y}_i|$
$\mathcal{N} \leftarrow$ The first $\alpha n$ decision variables in $N_{now}$
$\mathcal{F} \leftarrow \{x \| x \in X \wedge x \notin N\}$         ▷Fixed
$\mathcal{U} \leftarrow \{x \| x \in X \wedge x \in N\}$         ▷Unfixed
$\mathcal{X}' \leftarrow \text{SEARCH}(\mathcal{F}, \mathcal{U}, \mathcal{X})$
**Return:** $\mathcal{X}'$

---

where $\mathcal{P}_i$, $\mathcal{X}_i$ and $\hat{y}_i$ denotes the prediction loss, the value in the current solution and the predicted value of the $i$-th decision variable respectively.

---

**Algorithm 4** Neighborhood Crossover

---

**Input:** The set of decision variables $X$, the number of decision variables $n$, neighborhood $N_1, N_2$, neighborhood search solution $\mathcal{X}_1', \mathcal{X}_2'$
**Init:** Neighborhood crossover solution $\mathfrak{X} = \{\}$
$\mathcal{X}'' \leftarrow \{\}$
**for** $i = 1$ **to** $n$ **do**
    **if** The $i$-th decision variables in $N_1$ **then**
        $\mathcal{X}''[i] \leftarrow \mathcal{X}_1'[i]$
    **else**
        $\mathcal{X}''[i] \leftarrow \mathcal{X}_2'[i]$
    **end if**
**end for**
$\mathcal{F} \leftarrow X$         ▷Fixed
$\mathcal{U} \leftarrow \emptyset$         ▷Unfixed
$\mathcal{F}', \mathcal{U}' \leftarrow \text{REPAIR}(\mathcal{F}, \mathcal{U}, \mathcal{X}'')$
**if** $|\mathcal{U}'| \leq \alpha n$ **then**
    $\mathfrak{X} \leftarrow \text{SEARCH}(\mathcal{F}', \mathcal{U}', \mathcal{X}'')$
**end if**
**Return:** $\mathfrak{X}$

---

Given that the size of the neighborhood is constrained to at most $\alpha n$, there is a heightened risk of becoming trapped in local optima due to the limited radius of the neighborhood search. Consequently, neighborhood crossover plays a pivotal role. Algorithm 4 outlines the specific procedure for crossing two neighborhoods, denoted as $N_1$ and $N_2$.

## A.8 REPAIR ALGORITHM

---

**Algorithm 5** REPAIR Algorithm

---

**Input:** The set of fixed variables $\mathcal{F}$, the set of unfixed variables $\mathcal{U}$, the current solution $\mathcal{X}$
$\{A, b, l, u\} \leftarrow$ The coefficient of the given MILP
$n \leftarrow$ the number of decision variables
$m \leftarrow$ the number of constraints
**for** $i = 1$ **to** $m$ **do**
  $\mathcal{N} \leftarrow 0$
  **for** $j = 1$ **to** $n$ **do**
    **if** The $j$-th decision variable $\in \mathcal{F}$ **then**
      $\mathcal{N} \leftarrow \mathcal{N} + \mathcal{X}_j * A_{i,j}$
    **else**
      **if** $A_{ij} > 0$ **then**
        $\mathcal{N} \leftarrow \mathcal{N} + l_j * A_{i,j}$
      **end if**
      **if** $A_{ij} < 0$ **then**
        $\mathcal{N} \leftarrow \mathcal{N} + u_j * A_{i,j}$
      **end if**
    **end if**
  **end for**
  **if** $\mathcal{N} > b_i$ **then**
    **for** $j = 1$ **to** $n$ **do**
      **if** The $j$-th decision variable $\in \mathcal{F}$ **then**
        Remove the $j$-th decision variable from $\mathcal{F}$
        Append the $j$-th decision variable into $\mathcal{U}$
        $\mathcal{N} \leftarrow \mathcal{N} - \mathcal{X}_j * A_{i,j}$
        **if** $A_{ij} > 0$ **then**
          $\mathcal{N} \leftarrow \mathcal{N} + l_j * A_{i,j}$
        **end if**
        **if** $A_{ij} < 0$ **then**
          $\mathcal{N} \leftarrow \mathcal{N} + u_j * A_{i,j}$
        **end if**
        **if** $\mathcal{N} \leq b_i$ **then**
          **BREAK**
        **end if**
      **end if**
    **end for**
  **end if**
**end for**
**Return:** $\mathcal{F}, \mathcal{U}$

---

The initial solution search, neighborhood search and individual crossover with a lightweight optimizer for MILP are widely used in the proposed framework to improve the current solution, which can be written as the following.

$$
\begin{aligned}
\min_{x \notin \mathcal{F}} \ & c^T x \\
\text{subject to } & Ax \leq b, l \leq x \leq u, \\
& x_j \in \mathbb{Z}, j \in \mathbb{I}, \\
& x_i = \hat{x}_i, \forall x_i \in \mathcal{F},
\end{aligned}
\tag{16}
$$

where $\hat{x}_i$ represents the value of the $i$-th decision variable in the current solution, and $\mathcal{F}$ refers to the set of decision variables fixed to their current solution values. However, it is worth noting that

the Mixed-Integer MILP corresponding to Equation (16) may become infeasible, resulting in the failure of the current solution. To address this issue, we introduce a REPAIR Algorithm designed to examine and rectify constraints that are inherently infeasible by removing the fixation of certain illegal variables associated with these infeasible constraints.

In particular, when dealing with a given Mixed-Integer Linear Programming (MILP) problem alongside a set of fixed variables denoted as $\mathcal{F}$, the REPAIR algorithm systematically iterates through each constraint within the MILP. For each constraint under consideration, the algorithm assesses whether it is inevitably infeasible based on the upper and lower bounds of unfixed variables. If the algorithm determines that the constraint is indeed destined to be infeasible, it proceeds to release the fixation of specific decision variables associated with the constraint, aiming to restore its feasibility. The intricate steps of this process are elucidated in Algorithm 5.

### A.9 PROBLEM REDUCTION RATE

Regarding the chosen dimension reduction ratio ($\alpha$), it can indeed only be at an ($\mathcal{O}(1)$) level, and achieving a final subproblem significantly smaller than the original problem is not feasible. However, this has crucial practical implications. For mixed-integer programming problems, people typically resort to solvers. The scalability of solvers depends on both internal algorithmic settings and the user's machine configuration. In a given solver environment and computational resource constraints, individuals can only solve problems within a specific size limit, for example, those with ($k$) decision variables. Previous research on machine learning-based solutions for large-scale mixed-integer programming problems also often required solvers of the same size as the problem being tackled(Sonnerat et al., 2021; Wu et al., 2021).

In our work, we introduced a dimension reduction approach that simultaneously operates on the decision variable and constraint levels. In our test problems, when the number of decision variables reduces to 30% and the number of constraints reduces to 20% of the original problem, our framework can still generate high-quality feasible solutions. This implies that within a given solver environment and computational resource constraints, we can surpass the limits imposed by solving problems of specific sizes (e.g., with ($k$) decision variables) and efficiently solve larger-scale mixed-integer programming problems (e.g., with ($2k$) or ($3k$) decision variables). Therefore, proposing the dimension reduction ratio ($\alpha$) not only improves the efficiency of problem-solving but also serves as inspiration for overcoming physical resource constraints in solving even larger-scale problems.

It is worth notice that the choice of the decision variable reduction parameter $\alpha$ (which is set at approximately 30% based on empirical findings (Nair et al., 2020)) is fixed at the beginning and remains small-scale throughout the run; it does not change during the run to large-scale. For constraint reduction, to reduce the drawbacks of empirical settings, we adopt a progressive strategy. Initially, more constraint conditions are retained, and as optimization progresses, redundant constraints are automatically identified based on the current solution. The number of constraint conditions gradually decreases through the removal of redundant constraints. Practical experiments have shown that for most baseline tests, we can achieve a reduction of decision variables to 30% and constraints to 20% of the original problem size. The automatic selection of $\alpha$ is our next research goal, and we will explore learning algorithms to automatically determine the value of $\alpha$. It is worth mentioning that with the variation of $\alpha$, the final value is generally a unimodal function, effectively achieving the use of a certain scale of small-scale solvers to solve large-scale problems.

## B EXPERIMENTS DETAILS

### B.1 EXPERIMENTAL SETTINGS

All experiments are run on a machine with Intel Xeon Platinum 8375C @ 2.90GHz CPU and four NVIDIA TESLA V100(32G) GPU. Each scale of any Benchmark MILP is tested on five different instances, and the results shown are the average of the five results.

## B.2 BASELINES

In this paper, two types of baselines are employed. One type of the baselines is the latest ML-based optimization framework based on GNN&GBDT(Ye et al., 2023b). The other type of baselines is the state-of-the-art MILP solvers, including SCIP(4.3.0) (Achterberg, 2009) and Gurobi(10.0.1) (Achterberg, 2019). Their scale-constrained versions are used as the lightweight optimizer for both the proposed framework and the latest ML-based optimization framework.

In the comparative experiments with the advanced solver Gurobi (or SCIP), Gurobi is utilized in three instances: first, as the baseline solver; second, within the baseline machine learning framework GNN&GBDT, Gurobi with decision variable scale constraints is employed as a small-scale optimizer; third, in the proposed framework Light-MILPopt, Gurobi with decision variable scale constraints is used as a small-scale optimizer. In all these cases, Gurobi is used with default settings, without any parameter modifications, for reading and solving the problems. This approach of using solvers with default settings for fair comparisons is a common practice in prior works (Gasse et al., 2019; Nair et al., 2020; Sonnerat et al., 2021; Ye et al., 2023b; Wu et al., 2021).

## B.3 DATASET

For the four widely used NP-hard benchmark MILPs, the existing data set cannot meet such large-scale data requirements, so we use data generators to generate training and test data sets. Specifically, for the Maximum Independent Set problem (MIS) or Minimum Vertex Covering problem (MVC) with $n$ decision variables and $m$ constraints, we generate a random graph with $n$ nodes and $m$ edges to correspond to an MILP that meets the scale requirements. For the Set Covering problem (SC) with $n$ decision variables and $m$ constraints, we generate a random problem with $n$ items and $m$ sets where each set bid includes $4$ items. For the Mixed Integer Knapsack Set (MIKS) with $n$ decision variables and $m$ constraints, we generate a random problem with $n$ items and $m$ dimension where each dimension includes $4$ items and at least half of the items are items with integer constraints. For the optimal solution in the training data set, we use Gurobi to run for 8 hours to find the approximate optimal solution.

The decision variables and constraint scale of the one case study in the internet domain and four widely used NP-hard benchmark MILPs are shown in Table 3.

Table 3: The size of one real-world case study in the internet domain and four widely used NP-hard benchmark MILPs. SC denotes the Set Covering problem. MVC denotes the Minimum Vertex Covering problem. MIS denotes the Maximum Independent Set problem. MIKS denotes the Mixed Integer Knapsack Set problem. Case Study denotes the real-world case study.

| Problem | Scale | Number of Variables | Number of Constraints |
|---|---|---|---|
| SC | $SC_1$ | 200000 | 200000 |
| (Minimize) | $SC_2$ | 2000000 | 2000000 |
| MVC | $MVC_1$ | 100000 | 300000 |
| (Minimize) | $MVC_2$ | 1000000 | 3000000 |
| MIS | $MIS_1$ | 100000 | 300000 |
| (Maximize) | $MIS_2$ | 1000000 | 3000000 |
| MIKS | $MIKS_1$ | 200000 | 200000 |
| (Maximize) | $MIKS_2$ | 2000000 | 2000000 |
| Case Study (Maximize) | Case Study | 2040000 | 100003 |

## C ADDITIONAL EXPERIMENTAL RESULTS AND DISCUSSIONS

### C.1 COMPARISON OF CONSTRAINTS REDUCTION

To facilitate ablation experiments concerning constrained reduction, a pivotal innovation within Light-MILPopt, and to validate the efficacy of this technique, we conducted a comparative analysis. We evaluated the solution outcomes of our proposed frameworks under two conditions: with

Table 4: Comparison of the solving results of the Light-MILPopt optimization framework with constrains reduction and without constrains reduction for a fixed runtime. With-30%S means the proposed framework with the scale-limited versions of SCIP which limit the variable proportion $\alpha$ to 30% and with constrains reduction. Without-50%S means the proposed framework with the scale-limited versions of SCIP which limit the variable proportion $\alpha$ to 50% and without constrains reduction.

| | $SC_2$ | $MVC_2$ | $MIS_2$ | $MIKS_2$ |
|---|---|---|---|---|
| With-30%S | **166756.0** | **273014.6** | **227074.5** | **355887.6** |
| Without-30%S | 167169.5 | 292510.1 | 174894.4 | 354614.6 |
| With-50%S | **166966.9** | **269771.3** | **230278.1** | **357483.8** |
| Without-50%S | 197515.8 | 284936.2 | 190243.5 | 349676.3 |
| Time | 12000s | 8000s | 8000s | 12000s |

Table 5: The size of three Ultra-large-scale benchmark MILPs. SC represents Set Covering, MVC represents Minimum Vertex Cover, and MIS represents Maximum Independent Set.

| | $SC_3$ | $MVC_3$ | $MIS_3$ |
|---|---|---|---|
| **Number of Variables** | 20000000 | 10000000 | 10000000 |
| **Number of Constraints** | 20000000 | 30000000 | 30000000 |

and without constrained dimensionality reduction. This evaluation was performed on four standard large-scale Mixed-Integer Linear Programs (MILPs) under fixed solution time constraints. The results of these experiments are presented in Table 4. The findings from our experiments demonstrate that constrained reduction significantly enhances the performance of our framework across all test problems. It effectively improves the framework's ability to find solutions and boosts overall solving efficiency.

## C.2 FURTHURE ANALYSIS OF CONVERGENCE

To delve deeper into the solution performance and convergence capabilities of Light-MILPopt across varying problem sizes, we conducted a comprehensive analysis. We generated time-objective function plots for our proposed framework as well as baseline algorithms, utilizing Gurobi as a sub-solver, across a range of standard benchmark MILPs. The experimental results, depicted in Figure 6 and Figure 7, unequivocally demonstrate the superiority of our proposed framework. Regardless of the problem size, it consistently outperforms the baseline algorithm, producing higher-quality solutions within fixed time constraints. Furthermore, our framework exhibits convergence performance on par with that of the commercial solver Gurobi.

## C.3 COMPARISON ON ULTRA-LARGE-SCALE MILPs

We extend the generalization capabilities of the proposed Light-MILPopt method, trained on small-scale data with decision variables and constraints in the order of tens of thousands, to solve problems with decision variables and constraints in the order of tens of millions. The problem scales are illustrated in the Table 5, where SC represents Set Covering, MVC represents Minimum Vertex Cover, and MIS represents Maximum Independent Set.

The solution results are presented in the Table 6, where Ours-30%S and Ours-50% represent the results of the proposed framework using only 30% or 50% of the original problem scale for lightweight small-scale SCIP solving. SCIP indicates the results obtained directly using the baseline solver SCIP. It is evident that the proposed Light-MILPopt method maintains a significant advantage over the baseline solver, even when dealing with problem scales in the order of tens of millions for both decision variables and constraints.

## C.4 ANALYSIS OF GENERALIZATION CAPABILITIES

The proposed solution method for solving large-scale MILPs based on EGAT with half convolutions structure is notable for its use of the FENNEL-based problem partitioning strategy. This strategy

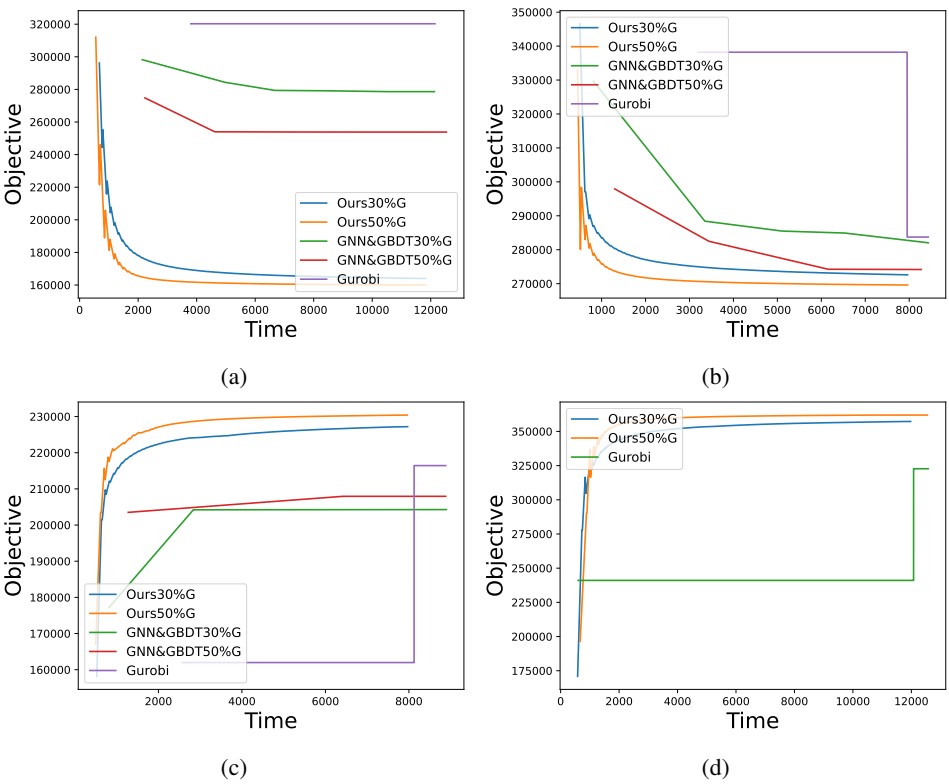

Figure 6: Time-objective figure for the large-scale benchmark MLIPs. (a) The minimized $SC_2$ problem. (b) The minimized $MVC_2$ problem. (c) The minimized $MIS_2$ problem. (d) The minimized $MIKS_2$ problem.

Table 6: Comparison of the solving results of the Light-MILPopt optimization framework with SCIP on three Ultra-large-scale benchmark MILPs. With-30%S means the proposed framework with the scale-limited versions of SCIP which limit the variable proportion $\alpha$ to 30%. With-50%S means the proposed framework with the scale-limited versions of SCIP which limit the variable proportion $\alpha$ to 50%.

|  | $SC_3$ | $MVC_3$ | $MIS_3$ |
|---|---|---|---|
| **Ours-30%G** | 1667157.94 | 2724414.73 | 2267990.75 |
| **Ours-50%G** | **1603278.85** | **2694126.35** | **2299504.56** |
| **Gurobi** | 3198747.63 | 2834161.28 | 2165906.72 |
| **Ours-30%S** | **1672097.50** | 2731152.61 | 2256644.32 |
| **Ours-50%S** | 2889696.49 | **2696953.27** | **2299950.04** |
| **SCIP** | 9190301.09 | 4909317.99 | 90750.01 |
| **Time** | 80000s | 80000s | 80000s |

decomposes large-scale MILPs into several smaller subproblems, predicting the optimal solution for each subproblem and concatenating them to obtain the initial feasible solution for the complete problem. This approach allows training and inference to be performed exclusively on smaller-scale problems. Therefore, for specific new optimization problems, it is not constrained by the decision variable and constraint quantities of existing training datasets, demonstrating robust generalization capabilities across different scales of new optimization problems. For instance, while the training dataset comprises small-scale data with decision variables and constraints in the order of tens of thousands, the testing dataset includes problems with decision variables and constraints ranging from hundreds of thousands to millions, as shown in Table 1 and Table 2.

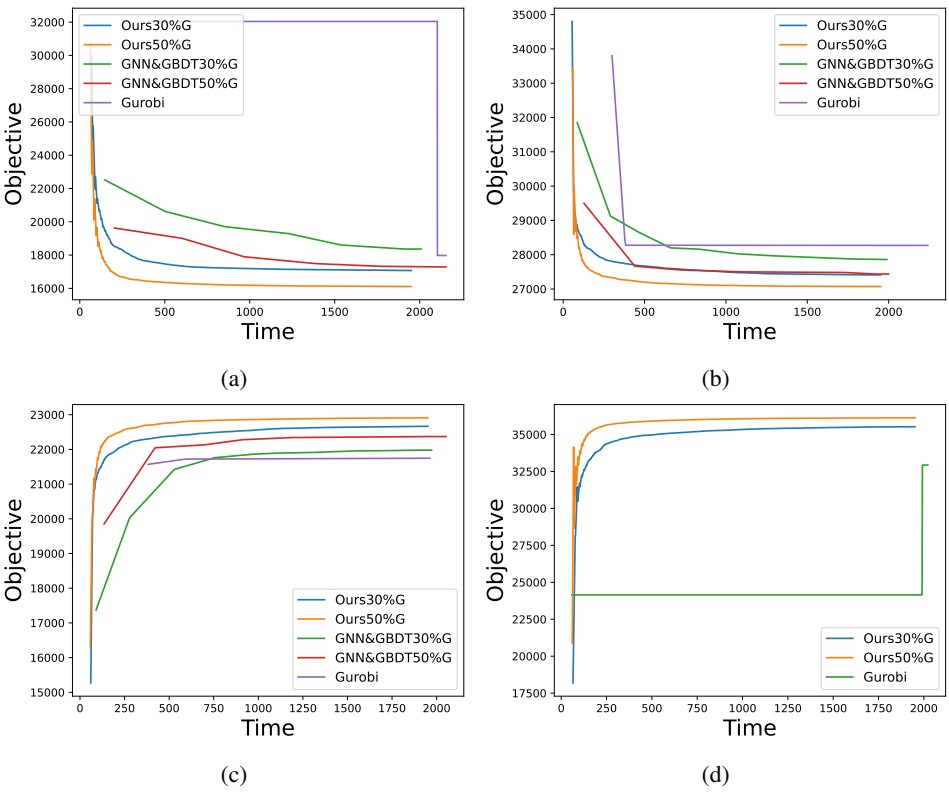

Figure 7: Time-objective figure for the medium-scale benchmark MLIPs. (a) The minimized $SC_1$ problem. (b) The minimized $MVC_1$ problem. (c) The minimized $MIS_1$ problem. (d) The minimized $MIKS_1$ problem.

Table 7: Comparison of initial solution prediction results applying only small-scale training data versus using a mixture of small-scale and medium-scale training data. Ours-30% means the proposed framework with variable proportion $\alpha$ to 30% using only small-scale training data. MIX-30% means the proposed framework with variable proportion $\alpha$ to 50% using a mixture of small-scale and medium-scale training data.

|  | $SC_1$ | $SC_2$ | $MVC_1$ | $MVC_2$ | $MIS_1$ | $MIS_2$ |
|---|---|---|---|---|---|---|
| **Ours-30%** | 29962.09 | 295981.16 | 34634.96 | 346677.69 | **15275.82** | **158352.06** |
| **MIX-30%** | **28882.44** | **290205.17** | **31788.13** | **318258.13** | 14577.61 | 152015.11 |
| **Ours-50%** | 30694.93 | 312419.34 | 33280.27 | 333192.51 | **16384.02** | **167161.47** |
| **MIX-50%** | **29710.68** | **300873.89** | **32424.85** | **323032.42** | 16044.14 | 162911.65 |

The proposed method also demonstrates adaptability to different scales of training data and uniform learning capabilities. Even when training data vary in scale, the model can be trained consistently and coherently. We supplement the results of training using a mixture of small-scale and medium-scale training data, shown in Table 7. Specifically, Ours represents the results of training with only small-scale training data, while Mix represents the results of training with the addition of medium-scale training data on top of small-scale training data. The results indicate that, in the case of SC and MVC problems, adding medium-scale training data enhances the generalization capability for initial solution prediction in large-scale data. Meanwhile, for MIS problems, the addition of training data of different scales yields results comparable to those without further improvement, suggesting saturation in the training data for the specific problem. Thus, even when using training data containing problems of different scales, the trained model can efficiently solve large-scale problems.

Table 8: Comparison of initial solution prediction results using confidence threshold versus using SelectiveNet. Ours-30% represents the initial feasible solution results obtained by reducing decision variables to 30% of the original problem using confidence threshold-based reduction, while SelectiveNet-50% represents the results obtained by reducing decision variables to 50% of the original problem using SelectiveNet.

|  | $SC_1$ | $SC_2$ | $MVC_1$ | $MVC_2$ | $MIS_1$ | $MIS_2$ |
|---|---|---|---|---|---|---|
| **Ours-30%** | 29962.09 | 295981.16 | **34634.96** | **346677.69** | **15275.82** | **158352.06** |
| **SelectiveNet-30%** | **25605.41** | **261728.93** | 49993.15 | 500171.37 | 13804.92 | 145147.94 |
| **Ours-50%** | 30694.93 | 312419.34 | **33280.27** | **333192.51** | **16384.02** | **167161.47** |
| **SelectiveNet-50%** | **25605.41** | **261728.93** | 49993.15 | 500171.37 | 13804.92 | 145147.94 |

Table 9: Comparison of the selection of network structure in initial feasible solution prediction. Var-30% represents predicting 70% of decision variables, and the remaining 30% are solved using a small-scale solver.

|  | **GNN&GBDT** | **GAT** | **EGAT** |
|---|---|---|---|
| **Var-30%** | 1817.3 | 1530.7 | **1872.3** |
| **Var-40%** | 1928.5 | 1681.8 | **2002.3** |
| **Var-50%** | 2007.0 | 1838.2 | **2067.9** |
| **Var-60%** | 2036.1 | 1987.8 | **2096.5** |

## C.5 ABLATION STUDIES

### C.5.1 VARIABLE REDUCTION METHODS

For the choice of decision variable reduction strategy, we compare two methods: SelectNet and confidence threshold-based reduction. The experimental results are presented in the Table 8. The results indicate that SelectiveNet selected in this paper has an advantage in one of the three problems and is noticeably inferior in the remaining two problems compared to the current method. Considering that SelectiveNet requires retraining the network for different reduction ratios, whereas the confidence threshold-based method requires training only once and can be used for different reduction ratios, we ultimately chose the confidence threshold-based selection method. Furthermore, further analysis revealed that the fixed selection threshold in SelectiveNet led to conservative model predictions. In subsequent experiments, we combined the strengths of SelectNet and confidence threshold-based methods, resulting in better overall results.

### C.5.2 NETWORK STRUCTURE SELECTION

In the model-based initial solution prediction module, classical methods often use graph convolutional neural networks (Nair et al., 2020; Gasse et al., 2019). However, these methods do not consider different correlations between neighborhoods. Therefore, Ding et al. (2020) introduced the GAT with an attention mechanism to capture correlations between points for better initial solution prediction. However, this GAT only updates node features and ignores the positive impact of edge feature updates on neighborhood aggregation. Therefore, we further introduced the EGAT (Gong & Cheng, 2019) with an edge update mechanism, combined with half convolutions layers for higher computational efficiency. We also compared the above methods and prediction methods based on GNN&GBDT (Ye et al., 2023b) in the preliminary exploration.

So we compare the selection of network structure in initial feasible solution prediction. In our earlier exploration, we tried the GNN&GBDT structure, the integrated GAT structure, and the integrated EGAT structure with multiple layers of half convolutions layers. We conducted comparative tests on small-scale maximization MILP problems, where Var-30% represents predicting 70% of decision variables, and the remaining 30% are solved using a small-scale solver. Higher values indicate better predictive performance. The results are shown in Table 9. It show that the current method, which adopts the EGAT method with multiple layers of half convolutions, has a clear advantage.

Table 10: Comparison of the proposed framework and Gurobi's initial solution generation. Ours-30% represents predicting 70% of decision variables, and the remaining 30% are solved using a scale-limited versions of Gurobi which limit the variable proportion $\alpha$ to 30%.

| | $SC_1$ | $SC_2$ | $MVC_1$ | $MVC_2$ | $MIS_1$ | $MIS_2$ |
|---|---|---|---|---|---|---|
| **Ours-30%** | **29962.09** | **295981.16** | 34634.96 | 346677.69 | 15275.82 | 158352.06 |
| **Ours-50%** | 30694.93 | 312419.34 | **33280.27** | **333192.51** | **16384.02** | **167161.47** |
| **Gurobi** | 32040.60 | 320272.92 | 33802.45 | 338198.32 | 16190.70 | 161973.05 |

Table 11: Comparison of initial solution generation time between the proposed framework and Gurobi.

| | $SC_1$ | $SC_2$ | $MVC_1$ | $MVC_2$ | $MIS_1$ | $MIS_2$ |
|---|---|---|---|---|---|---|
| **Ours-Time** | **68.88s** | **655.00s** | **53.81s** | **495.38s** | **49.60s** | **480.74s** |
| **Gurobi-Time** | 336.77s | 3399.88s | 375.37s | 2702.16s | 279.23s | 2742.89s |

### C.5.3 INITIAL SOLUTION GENERATION STRATEGIES

We compare the initial solution generation capabilities of the proposed method with commercial solver Gurobi on three standard benchmark problems, that is shown in table 10. The results show that, under the same generation time, the proposed method, aided by initial solution prediction, can obtain better initial feasible solutions than Gurobi.

Further analysis shown in table 11 reveals that our initial solution prediction method not only obtains better initial feasible solutions but also significantly improves efficiency compared to Gurobi. It requires only 20% of the time Gurobi needs to generate feasible solutions and achieves better-quality initial feasible solutions.

### C.6 ADDITIONAL INSTANCES FROM MIPLIB

To further validate the effectiveness of the proposed method in solving initial feasible solutions for complex real-world problems, we conducted tests on real-world problems SCP from MIPLIB. The results demonstrate that the proposed framework can effectively predict initial solutions for real-world problems and efficiently solve them. The table below presents experimental results.

Table 12: Comparison of objective value results with baselines under the same running time on SCP. Ours-30%S represents the results obtained by the proposed framework using a restricted version of SCIP with a solution size limited to 30% of the original problem, and Ours-50%G represents the results obtained using a restricted version of Gurobi with a solution size limited to 50% of the original problem.

| | scpm1 | scpn2 |
|---|---|---|
| **Ours-30%S** | **718.0** | **666.0** |
| **Ours-50%S** | 754.0 | 700.0 |
| **SCIP** | 807.0 | 19145.0 |
| **Ours-30%G** | **662.0** | 665.0 |
| **Ours-50%G** | 676.0 | **604.0** |
| **Gurobi** | 836.0 | 793.0 |
| **Time** | 2000s | 2000s |

## D ADDITIONAL INFORMATION ON MILPS

Mixed Integer Linear Programs (MILPs) are a type of problem in which the objective function is linear under several linear constraints, where some or all decision variables are restricted to take integer values, and the other decision variables are real numbers. Formally, an MILP has the form as the following.

$$\min_x c^T x, \text{subject to } Ax \leq b, l \leq x \leq u, x_i \in \mathbb{Z}, i \in \mathbb{I}, \tag{17}$$

where $x$ are the decision variables whose number is denoted by $n \in \mathbb{Z}$, with $l, u, c \in \mathbb{R}^n$ being their lower bound, upper bound and coefficient, respectively. $A \in \mathbb{R}^{m \times n}$ and $b \in \mathbb{R}^m$ denote the linear constraints. $\mathbb{I} \subseteq \{1, 2, \ldots, n\}$ is the index set of integer variables.

A solution is feasible for the MILP if decision variables $x \in \mathbb{R}^n$ satisfy all the constraints in Equation (17). An MILP is infeasible if there exists a feasible solution, while it is infeasible if there exists no solution that satisfies all of the constraints – in other words, if no feasible solution can be constructed (Guieu & Chinneck, 1999).

A feasible solution is optimal if it attains the minimum objective function value of the minimized MILP. An MILP is bounded if there exists an optimal solution, while unbounded if the objective function may be improved indefinitely without violating the constraints and bounds (Byrd et al., 1987).

