# OpenReview forum: "Light-MILPopt: Solving Large-scale Mixed Integer Linear Programs with Lightweight Optimizer and Small-scale Training Dataset"
_ICLR.cc/2024/Conference — ICLR 2024 poster_

### Official Review · Reviewer_RzQr · 2023-10-26

**Soundness:** 2 fair
**Presentation:** 2 fair
**Contribution:** 2 fair
**Rating:** 3
**Confidence:** 4

**Summary:**

This paper presents Light-MILPopt, a novel, lightweight optimization framework designed for large-scale Mixed Integer Linear Programs (MILPs). Conventional methods encounter difficulties, including high computational costs and complexities, when applied to large-scale MILPs. Light-MILPopt confronts these problems by adopting a four-stage strategy:

Problem Formulation: MILPs are depicted as bipartite graphs, and computational efficiency is improved through a graph partitioning algorithm.
Model-based Initial Solution Prediction: This stage involves using a specialized network to predict initial solutions for subproblems, necessitating less computational effort and a smaller, structurally coherent training dataset.
Problem Reduction: This innovative method selectively diminishes decision variables and constraints, thereby boosting the efficiency of problem reduction.
 Data-driven Optimization: This phase utilizes subgraph clustering and active constraint updating to steer the neighborhood search and optimization processes, iteratively enhancing solutions with limited computational demands.

Through exhaustive testing against large-scale benchmark MILPs and in real-world situations, the framework has proven its effectiveness. It surpasses current methods, emphasizing considerable reductions in computational requirements and improved problem-solving abilities. Light-MILPopt stands out particularly for its performance with smaller-scale training data and optimizers, establishing a new standard for the resource-efficient tackling of extensive MILPs. The contributions of this paper are crucial, offering a methodology that not only simplifies the computational procedure but also strengthens the strategy for addressing sophisticated, large-scale MILP difficulties.

**Strengths:**

This paper emphasizes the unique strengths of the Light-MILPopt framework for large-scale Mixed Integer Linear Programs (MILPs). Key highlights include its revolutionary four-stage process that enhances problem-solving efficiency and effectiveness, and its notable resource efficiency, being the first to tackle extensive MILPs with minimal computational resources. Light-MILPopt utilizes cutting-edge computational methods, effectively reduces problem dimensionality, and its efficacy is confirmed through rigorous testing, outperforming existing models. The framework significantly contributes to future MILP endeavors, particularly in efficiently solving substantial problems with limited resources.

**Weaknesses:**

The Light-MILPopt framework, despite its innovative approach to large-scale Mixed Integer Linear Programs (MILPs), faces critical limitations, including questions of scalability and generalization due to its dependence on small-scale training data. The complexity inherent in its advanced problem formulation and division techniques may hinder practical application, necessitating specialized expertise. The framework's simplification strategies in problem reduction could potentially neglect essential problem aspects, compromising solution integrity. Additionally, its heavy reliance on model-based predictions introduces risks of inaccuracies, and the lack of detailed validation experiments obscures comprehensive performance assessment. Finally, comparisons with advanced solvers appear limited to certain metrics, calling for a more exhaustive evaluation. Overall, these issues indicate the need for cautious implementation and enhanced development for broader applicability.

**Questions:**

MILP is an NP-hard problem, and, in the worst case, finding an optimal solution in polynomial time is considered impossible (though this has not yet been proven). The No Free Lunch theorem posits that no "universal" supervised machine learning model or search/optimization algorithm exists that can efficiently solve every problem (theoretically, it's unfeasible). However, assuming a prior distribution of problem generation, it becomes possible to efficiently solve specific instances with an approach like the one used in this study. Nonetheless, a substantial gap exists, limiting the effectiveness of such methods. The explanation of the limitations and constraints of this method in this context is inadequate.

Answering all of the following questions in a necessary and sufficient manner would be an onerous task for the authors; therefore, please respond as comprehensively as possible at this juncture.

1: Representativeness and diversity of the dataset: Is the dataset chosen for benchmarking genuinely representative of the diverse spectrum of large-scale MILPs in real-world scenarios? How does the framework handle MILPs with varying attributes or those from different industries? While problems like Set Covering, Minimum Vertex Cover, Maximum Independent Set, and Mixed Integer Knapsack Set are appropriate for this method's application due to their definable problem structure, its effectiveness may be restricted for more complex practical problems. Consequently, using MIPLIB (https://miplib.zib.de/), where issues are meticulously curated for MIP benchmarking, is advisable.

2: Generalizability of results: Considering that Light-MILPopt trains on merely 1% of the large benchmark MILPs, how effectively does the framework generalize to unfamiliar or more intricate MILPs? Is there proof of steady performance across a broader problem set?

3: Fairness and consistency of benchmarks: Were the tests executed under uniform and fair conditions for all methods? Gurobi itself has numerous parameter settings, potentially leading to disputes from Gurobi's developers' standpoint.

4: Detailed comparative analysis: While the summary underscores Light-MILPopt's superior performance relative to other solvers, it doesn't delve into why specific solvers underperform. Are there fundamental inefficiencies or constraints within the baseline solvers leading to subpar outcomes?

5: Robustness and Efficiency: How does Light-MILPopt manage issues of diverse sizes and complexities? Does efficiency wane as the problem's size and intricacy escalate? What are the trade-offs between computational time and solution precision?

6: Solution Quality and Optimality: Light-MILPopt is said to procure better outcomes in a shorter period, but how do these solutions compare to the global optimum? Is there a compromise in solution quality, risking a mere local optimum?

7: Convergence Analysis: While convergence performance is broached, are scenarios where Light-MILPopt fails to converge efficiently identified? Comprehending its conduct under unfavorable conditions is also crucial.

8: Training scalability: Given the framework's high efficacy with sparse training data, has there been an exploration of how augmenting training data influences performance? Is there a threshold where the benefits diminish, or does efficiency perpetually enhance with additional data?

9: Analysis of real-world applications: The study incorporates real-world instances, but is this scrutiny exhaustive? Are the distinct constraints and hurdles present in real-world scenarios considered, and how does the system accommodate them?

10: In (1) on page 2, the definition is not MILP (Mixed Integer Linear Programming Problem) but ILP (Integer Linear Programming Problem)

---

> ### Author Response · Authors · 2023-11-22
>
> Thank you for your attention and recognition of our work, especially your acknowledgment of our efficient use of resources to address large-scale problems. Additionally, we appreciate the constructive questions you've raised, as they will undoubtedly contribute to further refining our paper and advancing the development of the field. Below, we provide detailed responses to your questions.
>
> - $\textbf{Q1:}$ Representativeness and diversity of the dataset: Is the dataset chosen for benchmarking genuinely representative of the diverse spectrum of large-scale MILPs in real-world scenarios? How does the framework handle MILPs with varying attributes or those from different industries? While problems like Set Covering, Minimum Vertex Cover, Maximum Independent Set, and Mixed Integer Knapsack Set are appropriate for this method's application due to their definable problem structure, its effectiveness may be restricted for more complex practical problems. Consequently, using MIPLIB (https://miplib.zib.de/), where issues are meticulously curated for MIP benchmarking, is advisable.
>
>   $\textbf{A1:}$ Thank you for your questions!
>
>   1. This paper aims to solve large-scale MILP problems using small-scale training data and lightweight solvers. We initially tested the proposed framework on four standard MILP problems, a common practice in prior works on machine learning-based mixed-integer programming [4,5,6]. Additionally, to further validate the framework's effectiveness, we conducted tests on a real-world problem in the Internet domain, as shown in Tables 1 and 2 in the main text.
>
>   2. The framework's solving approach is designed for problems with the same structure and mathematical properties. It involves unified training, prediction, and neighborhood search to obtain optimized solutions. For problems with different structures or mathematical properties, a secondary training process is needed to construct a new feasible solution prediction model tailored to the specific problem. For example, if the original problem involved training for a minimum point cover, and a new problem arises requiring a minimum set cover, a new feasible solution prediction model would need to be trained.
>
>   3. We appreciate your suggestion, and we have added more problems from MIPLIB to the experiments. Since MIPLIB problems are mostly not classified, and many of the large-scale problems do not have isomorphic counterparts, they are not suitable for machine learning-based solving methods that require isomorphic training data. Due to time constraints, we conducted training and testing on the large-scale MILP problem "scp," and the results are presented in the table below. Ours-30%S represents the results obtained by the proposed framework using a scaled-down version of SCIP with a constraint of 30% of the original problem size, and Ours-50%G represents the results using a scaled-down version of Gurobi with a constraint of 50% of the original problem size.
>
>      |               |   scpm1   |   scpn2   |
>      | :-----------: | :-------: | :-------: |
>      | **Ours-30%S** | **718.0** | **666.0** |
>      | **Ours-50%S** |   754.0   |   700.0   |
>      |   **SCIP**    |   807.0   |  19145.0  |
>      | **Ours-30%G** | **662.0** |   665.0   |
>      | **Ours-50%G** |   676.0   | **604.0** |
>      |  **Gurobi**   |   836.0   |   793.0   |
>      |   **Time**    |   2000s   |   2000s   |
>
>      The results demonstrate that the proposed framework can effectively predict initial feasible solutions and achieve better solving performance than SCIP and Gurobi within the specified running time. We plan to conduct additional experiments on more diverse problems in the future.
>
>   4. For more complex problems, in the model-based initial solution prediction module, classical methods often use graph convolutional neural networks [2,6]. However, these methods do not consider different correlations between neighborhoods. Thus, Ding et al. [7] introduced GAT with an attention mechanism to capture correlations between points for better initial solution prediction. However, GAT can only update node features and neglects the positive effects of edge feature updates on neighborhood aggregation. Therefore, we further introduced EGAT with an edge update mechanism, combined with a half-convolution structure to achieve higher computational efficiency. However, relying solely on neural networks for initial solution prediction is often unreliable. Therefore, this paper introduces a Repair strategy (see Appendix Algorithm 5), which automatically identifies decision variables violating constraints, predicts and uses a small-scale solver to obtain feasible predictions, ultimately obtaining an initial feasible solution, thereby maximizing the probability of avoiding problems where an initial feasible solution cannot be found.

---

> ### Author Response · Authors · 2023-11-22
>
> - $\textbf{Q2:}$ Generalizability of results: Considering that Light-MILPopt trains on merely 1% of the large benchmark MILPs, how effectively does the framework generalize to unfamiliar or more intricate MILPs? Is there proof of steady performance across a broader problem set?
>
>   $\textbf{A2:}$ Thank you for your response!
>
>   This paper focuses on optimizing solutions for MILP problems with similar or identical mathematical structures. It employs a small-sample supervised learning strategy to predict initial feasible solutions and utilizes neighborhood search methods to obtain the final optimized solution. The framework's solving approach is designed for problems with the same structure and mathematical properties. It involves unified training, prediction, and neighborhood search to obtain optimized solutions. Additionally, it can efficiently handle large-scale problems using only small-scale training data. For problems with different properties, a secondary training process is required to construct a new feasible solution prediction model tailored to the specific problem. For example, problems with different scales of minimum point cover can be trained together. However, if the goal shifts to solving minimum set cover problems, a new feasible solution prediction model needs to be retrained.
>
> - $\textbf{Q3:}$ Fairness and consistency of benchmarks: Were the tests executed under uniform and fair conditions for all methods? Gurobi itself has numerous parameter settings, potentially leading to disputes from Gurobi's developers' standpoint.
>
>   $\textbf{A3:}$  Certainly, all experiments and tests conducted in this paper were performed in a standardized and fair environment. This encompasses a consistent hardware platform and software environment. During experiments, whenever Gurobi was invoked for comparative testing, the default recommended parameter settings of Gurobi were used.
>
> - $\textbf{Q4:}$ Detailed comparative analysis: While the summary underscores Light-MILPopt's superior performance relative to other solvers, it doesn't delve into why specific solvers underperform. Are there fundamental inefficiencies or constraints within the baseline solvers leading to subpar outcomes?
>
>   $\textbf{A4:}$ Thank you for your question!
>
>   1. According to the experimental results in Table 1 and Table 2 in the main text, we can observe that specific solvers (SCIP/Gurobi) perform poorly on solving super-large-scale problems. The analysis for this is as follows: as depicted in Figure 5 in the main text and Figures 6 and 7 in the appendix, solvers based on the Branch and Bound (B&B) method exhibit an exponential increase in search space for super-large-scale problems. This leads to a significant decrease in algorithm efficiency and an increase in complexity, resulting in deficiencies in both solving performance and effectiveness.
>
>   2. We propose a lightweight framework, Light-MILPopt, for optimizing large-scale mixed-integer linear programming (MILP) problems. Accurate initial feasible solution prediction and efficient problem dimensionality reduction are crucial for achieving efficient problem solving. The model-based initial solution prediction module utilizes EGAT (Edge-enhanced Graph Attention Network) with half convolutions layers for learning. The problem dimensionality reduction module employs confidence-based and KNN (K-Nearest Neighbors) algorithms to reduce the dimensionality of decision variables and constraints, respectively. Based on the predicted initial solutions, iterative improvements are made to the reduced subproblems, avoiding the issue of an excessively large search space for super-large-scale problems and thus achieving efficient problem solving.

---

> ### Author Response · Authors · 2023-11-22
>
> $\textbf{Q5:}$ Robustness and Efficiency: How does Light-MILPopt manage issues of diverse sizes and complexities? Does efficiency wane as the problem's size and intricacy escalate? What are the trade-offs between computational time and solution precision?
>
> $\textbf{A5:}$ Thank you for your question!
>
> 1. We propose a lightweight framework, Light-MILPopt, for optimizing large-scale mixed-integer linear programming (MILP) problems, where accurate initial feasible solution prediction and efficient problem dimensionality reduction are crucial for efficient problem solving. The model-based initial solution prediction module employs EGAT (Edge-enhanced Graph Attention Network) with haf convolutions layers for learning. The problem dimensionality reduction module uses confidence-based and KNN (K-Nearest Neighbors) algorithms to reduce the dimensionality of decision variables and constraints, respectively. Based on initial solution prediction and problem dimensionality reduction, our proposed algorithm can efficiently solve super-large-scale problems even in low-dimensional spaces.
>
> 2. As shown in the results of Table 1 and Table 2 in the main text, regardless of the number of decision variables and constraints being in the order of tens of thousands or millions, the efficiency of the proposed framework surpasses baseline solving algorithms comprehensively. This indicates that even as the problem size increases, our solution efficiency and effectiveness do not decrease. This is further confirmed in the latest MILP problems with dimensions in the order of tens of millions, demonstrating the robustness of our framework. The problem sizes are shown in the table below, where SC represents Set Covering, MVC represents Minimum Vertex Cover, and MIS represents Maximum Independent Set.
>
>    |                           |  SC$_3$  | MVC$_3$  | MIS$_3$  |
>    | :-----------------------: | :------: | :------: | :------: |
>    |  **Number of Variables**  | 20000000 | 10000000 | 10000000 |
>    | **Number of Constraints** | 20000000 | 30000000 | 30000000 |
>
>    The solution results are shown in the table below, where Ours-30%S and Ours-50% represent the results of the proposed framework using only 30% or 50% of the original problem size with the lightweight SCIP solver, respectively. SCIP represents the result of directly using the baseline solver SCIP for comparison. It is evident that the proposed Light-MILPopt method maintains a significant advantage even when the number of decision variables and constraints is in the order of tens of millions.
>
>    |               |     SC$_3$     |    MVC$_3$     |    MIS$_3$     |
>    | :-----------: | :------------: | :------------: | :------------: |
>    | **Ours-30%S** | **1672097.50** |   2731152.61   |   2256644.32   |
>    | **Ours-50%S** |   2889696.49   | **2696953.27** | **2299950.04** |
>    |   **SCIP**    |   9190301.09   |   4909317.99   |    90750.01    |
>    |   **Time**    |     80000s     |     80000s     |     80000s     |
>
> 3. To balance computation time and solution effectiveness, we introduce the dimensionality reduction factor, Alpha. Specifically, before solving, we set the decision variable reduction factor Alpha, indicating the use of a lightweight solver with solving capabilities not exceeding Alpha% of the original problem size for solving large-scale problems.
>
> 4. The choice of the decision variable reduction parameter Alpha (which is set at approximately 30% based on empirical findings [1]) is fixed at the beginning and remains small-scale throughout the run; it does not change during the run to large-scale. For constraint reduction, to reduce the drawbacks of empirical settings, we adopt a progressive strategy. Initially, more constraint conditions are retained, and as optimization progresses, redundant constraints are automatically identified based on the current solution. The number of constraint conditions gradually decreases through the removal of redundant constraints. Practical experiments have shown that for most baseline tests, we can achieve a reduction of decision variables to 30% and constraints to 20% of the original problem size. The automatic selection of Alpha is our next research goal, and we will explore learning algorithms to automatically determine the value of Alpha. It is worth mentioning that with the variation of Alpha, the final value is generally a unimodal function, effectively achieving the use of a certain scale of small-scale solvers to solve large-scale problems.

---

> ### Author Response · Authors · 2023-11-22
>
> - $\textbf{Q6:}$  Solution Quality and Optimality: Light-MILPopt is said to procure better outcomes in a shorter period, but how do these solutions compare to the global optimum? Is there a compromise in solution quality, risking a mere local optimum?
>
>   $\textbf{A6:}$ Thank you for your response!
>
>   The framework proposed in this paper is essentially a heuristic method framework. Heuristic search methods can obtain better results than state-of-the-art solvers and solving frameworks in a fixed, limited time, demonstrating clear advantages in situations with finite computational resources. However, there is a potential drawback of getting stuck in local optima. To overcome this issue, the paper explores the combination of neighborhood search and neighborhood crossover, incorporating prior knowledge from the partitioning of large-scale problems and the update of the active constraint set.
>
>   To validate the superiority of the proposed framework in terms of solution quality and efficiency, two types of experiments were conducted in the paper. The first type of experiment involved solving with nearly unlimited solving time. As seen in Experiment 2 (Table 2), the optimization efficiency of our proposed method surpasses the optimal solutions obtained by solvers. The second type of experiment focused on optimizing solving performance within a limited time. As shown in Experiment 1 (Table 1), within short optimization cycles, our proposed method outperforms optimization solvers such as Gurobi and SCIP on all test problems. Through both types of experiments, it is evident that on the experimental dataset, the proposed framework exceeds existing optimization solvers in both solution effectiveness and solving efficiency.
>
> - $\textbf{Q7:}$  Convergence Analysis: While convergence performance is broached, are scenarios where Light-MILPopt fails to converge efficiently identified? Comprehending its conduct under unfavorable conditions is also crucial.
>
>   $\textbf{A7:}$ Thank you for your questions!
>
>   1. Building upon the foundation of initial feasible solution prediction and problem dimensionality reduction, this paper employs an iterative improvement strategy involving neighborhood search and crossover. It utilizes a lightweight, small-scale solver to solve the reduced-dimensional subproblems iteratively, aiming to refine the current solution.
>   2. Neighborhood search fundamentally involves solving subproblems to enhance specific components of the current solution. It is akin to the method of large neighborhood hill climbing, progressively approaching an approximate optimal solution without degradation at each step. This ensures the convergence of our framework along a straight path.
>   3. Figures 5 in the main paper and Figures 5 and 6 in the appendix validate that the improvement methods based on neighborhood search progressively converge toward an optimized solution. The evidence indicates that the process is convergent and avoids non-convergence issues.

---

> ### Author Response · Authors · 2023-11-22
>
> - $\textbf{Q8:}$ Training scalability: Given the framework's high efficacy with sparse training data, has there been an exploration of how augmenting training data influences performance? Is there a threshold where the benefits diminish, or does efficiency perpetually enhance with additional data?
>
>   $\textbf{A8:}$ Thank you for your questions!
>
>   1. In real-life scenarios, when dealing with large-scale optimization problems, there are often only a limited amount of small-scale data available for training initial feasible solution prediction. The core work of this paper involves partitioning large-scale problems based on FENNEL graph partitioning to predict initial feasible solutions using only small-scale training data.
>
>   2. Conversely, for certain problem types, if there exists a larger training dataset, the framework proposed in this paper can be further improved. We have supplemented the results of training with a mixture of small-scale and medium-scale training data. Here, "Ours" represents the results trained solely on small-scale data, and "Mix" represents the results with the addition of medium-scale training data. The results show that, especially in the SC and MVC problems, adding medium-scale training data not only does not degrade predictive performance but also enhances the generalization ability for predicting initial solutions in large-scale data.
>
>      |              |    SC$_1$    |    SC$_2$     |   MVC$_1$    |    MVC$_2$    |   MIS$_1$    |    MIS$_2$    |
>      | ------------ | :----------: | :-----------: | :----------: | :-----------: | :----------: | :-----------: |
>      | **Ours-30%** |   29962.09   |   295981.16   |   34634.96   |   346677.69   | **15275.82** | **158352.06** |
>      | **Mix-30%**  | **28882.44** | **290205.17** | **31788.13** | **318258.13** |   14577.61   |   152015.11   |
>      | **Ours-50%** |   30694.93   |   312419.34   |   33280.27   |   333192.51   | **16384.02** | **167161.47** |
>      | **Mix-50%**  | **29710.68** | **300873.89** | **32424.85** | **323032.42** |   16044.14   |   162911.65   |
>
>   3. For standard baseline problems, we explored the use of larger datasets containing various scales of training data. From the table above, although there is improvement in some problems when new data is added, there is no significant improvement compared to the existing initial solution prediction. The experimental results indicate that, under existing conditions, the training data has reached a threshold, ensuring the quality of predicted solutions. In the MIS problem, adding training data of different scales also resulted in similar outcomes to the original, suggesting that the training data has saturated, and the threshold depends on the problem attributes.
>
> - $\textbf{Q9:}$ Analysis of real-world applications: The study incorporates real-world instances, but is this scrutiny exhaustive? Are the distinct constraints and hurdles present in real-world scenarios considered, and how does the system accommodate them?
>
>   $\textbf{A9:}$  Thank you for your question!
>
>   The case study discussed in this paper originates from a collaborative project with one of the world's largest O2O platforms, focusing on optimization and scheduling problems. The problem's description, definition, and data are derived from real-world applications on the platform. The formalization of the problem in the context of real-world scenarios was meticulously discussed and agreed upon by both parties, ensuring accuracy in the definition, including decision variables, constraints, and the objective function. Our solution strategy has been validated and approved by our collaborative partner, providing a viable solution for addressing practical optimization problems.
>
> - $\textbf{Q10:}$  In (1) on page 2, the definition is not MILP (Mixed Integer Linear Programming Problem) but ILP (Integer Linear Programming Problem)
>
>   $\textbf{A10:}$ Thank you for your question!
>
>   The MILP problem formulation defined in this paper is consistent with "Solving mixed integer programs using neural networks."[1] As of November 21, 2023, this paper has been cited 173 times. The definition method is also consistent with other references [2,3].

---

> > ### Comment · Reviewer_RzQr · 2023-11-22
> >
> > The essence of Q10 is not what you mentioned. In the case of MILP (Mixed Integer Linear Programming), there are situations where some variables are integers and others are real numbers. The correct definition is that MILP involves some variables being integers while the remaining variables are real numbers. Please take a look at page 2, equation (1) of the following paper for more details.
> > https://ris.utwente.nl/ws/portalfiles/portal/249527981/scipopt_60.pdf

---

> > > ### Comment · Reviewer_RzQr · 2023-11-22
> > >
> > > Regarding A3, what is the basis for saying as follows? In this method, hasn't there been any tuning of hyperparameters or similar aspects? If there has been, how can it be deemed a fair comparison when Gurobi is using default parameters, which are not tuned specifically for this instance?
> > > > A3. Certainly, all experiments and tests conducted in this paper were performed in a standardized and fair environment. This encompasses a consistent hardware platform

---

> > > > ### Comment · Reviewer_RzQr · 2023-11-22
> > > >
> > > > Regarding Q9, it seems that my intention in the question may not have been clearly conveyed, so I will rephrase it. I believe this method, like scp, excels in a predefined format. However, when considering practical problems, the constraints and the objective function become more complex, diverging significantly from the original scp formulation. In such cases, it might essentially become a general MILP problem. Can this method still be effective in such versatile problems? Looking at the response in A1 about MIPLIB, it seems challenging.

---

> > > > > ### Comment · Reviewer_RzQr · 2023-11-22
> > > > >
> > > > > Overall, if the formulation patterns are predetermined and their features can be extracted, I find this approach very interesting. However, on the other hand, it can also be seen as a very limited approach. Therefore, the following doubts of mine still remain.
> > > > > >MILP is an NP-hard problem, and, in the worst case, finding an optimal solution in polynomial time is considered impossible (though this has not yet been proven). The No Free Lunch theorem posits that no "universal" supervised machine learning model or search/optimization algorithm exists that can efficiently solve every problem (theoretically, it's unfeasible). However, assuming a prior distribution of problem generation, it becomes possible to efficiently solve specific instances with an approach like the one used in this study. Nonetheless, a substantial gap exists, limiting the effectiveness of such methods. The explanation of the limitations and constraints of this method in this context is inadequate.

---

> > > > > > ### Author Response · Authors · 2023-11-22
> > > > > > **Thank you very much for your response**
> > > > > >
> > > > > > Thank you very much for your response! We appreciate your positive feedback on our approach. We understand your concerns, and we'll provide detailed responses to your supplementary questions below. This will offer a more specific overview of the strengths and limitations of machine learning-based mixed-integer programming solving methods and relevant information about our proposed framework, Light-MILPopt.
> > > > > >
> > > > > > - $\textbf{Q11}:$ The essence of Q10 is not what you mentioned. In the case of MILP (Mixed Integer Linear Programming), there are situations where some variables are integers and others are real numbers. The correct definition is that MILP involves some variables being integers while the remaining variables are real numbers. Please take a look at page 2, equation (1) of the following paper for more details. https://ris.utwente.nl/ws/portalfiles/portal/249527981/scipopt_60.pdf
> > > > > >
> > > > > >   $\textbf{A11}$：Thank you for your question. In fact, the definition of mixed integer linear programming (MILP) problems in the references you provided is consistent with that in our manuscript. In the definition you presented, the MIP problem is formulated as follows,
> > > > > >   \begin{equation}
> > > > > >     \begin{aligned}
> > > > > >   min \space &c^Tx \\\\
> > > > > >   s.t \space &Ax \ge b, \\\\
> > > > > >   & l_i \le x_i \le u_i, \forall i \in \mathcal{N}, \\\\
> > > > > >   & x_i \in  \mathbb{Z}, \forall i \in \mathcal{I},
> > > > > >     \end{aligned}
> > > > > >   \end{equation}
> > > > > >   where $c \in \mathbb{R}^n$, $A \in \mathbb{R}^{m*n}$, $b \in \mathbb{R}^m$, $l, u \in \overline{\mathbb{R}}^n$, and the index set of integer variables $\mathcal{I} \subseteq \mathcal{N} := \{1, \dots, n\}$. This implies that for $i \in \mathcal{I}$, $x_i \in \mathbb{Z}$, and for $i \notin \mathcal{I}$， $x_i \in \mathbb{R}$.
> > > > > >
> > > > > >   However, in Equation (1) on page two of our manuscript, we noticed during the review that we modified $\mathbb{Z}^n$ to  $\mathbb{Z}$, while keeping the rest of the definition unchanged. Therefore, the formulation of the integer programming problem in our manuscript is as follows:
> > > > > >   \begin{equation}
> > > > > >     \begin{aligned}
> > > > > >   \mathop {\min}\limits_x \space &c^Tx\\\\
> > > > > >   \text{subject to} \space &Ax \ge b, \\\\
> > > > > >   & l \le x \le u, \\\\
> > > > > >   & x_i \in  \mathbb{Z},  i \in \mathcal{I}.
> > > > > >   \end{aligned}
> > > > > >   \end{equation}
> > > > > >   In the same paragraph, we mention that $\mathcal{I} \subseteq \{1, \dots, n\}$ is the index set of integer variables. This statement implies that for $i \in \mathcal{I}$, $x_i \in \mathbb{Z}$; while for $i \notin \mathcal{I}$, $x_i \in \mathbb{R}$. This satisfies your requirement of specifying that some decision variables are restricted to integers while the remaining variables are real numbers. We appreciate your question, as it prompts us to conduct a thorough review and make necessary revisions to ensure the accuracy of the manuscript.
> > > > > >
> > > > > > - $\textbf{Q12}:$ Regarding A3, what is the basis for saying as follows? In this method, hasn't there been any tuning of hyperparameters or similar aspects? If there has been, how can it be deemed a fair comparison when Gurobi is using default parameters, which are not tuned specifically for this instance?
> > > > > >
> > > > > >   $\textbf{A12}:$  Thank you for your perspective. In the comparative experiments with Gurobi, Gurobi is utilized in three instances: first, as the baseline solver; second, within the baseline machine learning framework GNN&GBDT, Gurobi with decision variable scale constraints is employed as a small-scale optimizer; third, in the proposed framework Light-MILPopt, Gurobi with decision variable scale constraints is used as a small-scale optimizer. In all these cases, Gurobi is used with default settings, without any parameter modifications, for reading and solving the problems. This approach of using solvers with default settings for fair comparisons is a common practice in prior works [1,2,3,4].

---

> > > > > > ### Author Response · Authors · 2023-11-22
> > > > > > **Thank you very much for your response**
> > > > > >
> > > > > > - $\textbf{Q13}:$  Regarding Q9, it seems that my intention in the question may not have been clearly conveyed, so I will rephrase it. I believe this method, like scp, excels in a predefined format. However, when considering practical problems, the constraints and the objective function become more complex, diverging significantly from the original scp formulation. In such cases, it might essentially become a general MILP problem. Can this method still be effective in such versatile problems? Looking at the response in A1 about MIPLIB, it seems challenging.
> > > > > >
> > > > > >   $\textbf{A13}:$ Thank you for your inquiry. In the introduction section of the main text, we focus on "a large number of homogeneous MILPs with similar combinatorial structures need to be solved simultaneously." Therefore, our proposed framework does not assume the specific form of the problem or the distribution of its mathematical formulation. It automatically learns the structure and distribution of isomorphic problems from training data composed of small-scale problems with similar mathematical structures. This allows it to predict initial solutions to address the cold-start problem that traditional methods still face when dealing with a large number of homogeneous problems. Consequently, there should not be a significant mismatch between the training and testing data.
> > > > > >
> > > > > >   In fact, the majority of methods for solving mixed-integer programming problems based on machine learning require separate training for each specific problem [5,6]. The models trained in this manner often outperform solvers on specific problems. However, for certain practical problems where constraints and objective functions vary widely and lack homogenous structures, machine learning-based approaches may no longer be applicable. In such cases, traditional solvers or classical solving algorithms need to be employed, starting from scratch each time.
> > > > > >
> > > > > >   We observe that both direct solvers and machine learning-based methods have their advantages and disadvantages, each suitable for different types of problems. Therefore, in our future work, we plan to heed your advice and attempt to integrate these two methods. This will involve using an automatic discriminator to identify the training environment of the problem at hand and automatically selecting the most suitable solving method. This integration aims to leverage the strengths of both approaches and efficiently solve a broader range of mixed-integer programming problems.

---

> > > > > > ### Author Response · Authors · 2023-11-22
> > > > > > **Thank you very much for your response**
> > > > > >
> > > > > > - $\textbf{Q14}:$​Overall, if the formulation patterns are predetermined and their features can be extracted, I find this approach very interesting. However, on the other hand, it can also be seen as a very limited approach. Therefore, the following doubts of mine still remain.
> > > > > >
> > > > > >   > MILP is an NP-hard problem, and, in the worst case, finding an optimal solution in polynomial time is considered impossible (though this has not yet been proven). The No Free Lunch theorem posits that no "universal" supervised machine learning model or search/optimization algorithm exists that can efficiently solve every problem (theoretically, it's unfeasible). However, assuming a prior distribution of problem generation, it becomes possible to efficiently solve specific instances with an approach like the one used in this study. Nonetheless, a substantial gap exists, limiting the effectiveness of such methods. The explanation of the limitations and constraints of this method in this context is inadequate.
> > > > > >
> > > > > >   $\textbf{A14}:$ Thank you for your insights. You are correct in stating that MILP is an NP-hard problem. Given current technology, proposing a polynomial-time algorithm to solve any mixed-integer programming problem is nearly impossible. There is also no "universal" supervised machine learning model or search/optimization algorithm that can efficiently solve all problems. This paper focuses on the common real-world scenario mentioned in the introduction: "a large number of homogeneous MILPs with similar combinatorial structures need to be solved simultaneously." In such cases, our approach, without assuming a specific problem distribution in advance, learns the structure and distribution of problems from training data with similar data structures to the test problems. This facilitates predicting solutions to address the cold-start problem.
> > > > > >
> > > > > >   While it's true that machine learning-based methods of this kind require retraining for each problem type, many companies or organizations deal with structurally similar problems daily (such as the case study we collaborated on). In this application scenario, traditional methods like solvers start from scratch every day and cannot effectively leverage prior knowledge gained from previous problem-solving. Machine learning-based methods, on the other hand, can effectively utilize experiential knowledge from past problem-solving, addressing the cold-start problem and playing an irreplaceable role in this context.
> > > > > >
> > > > > >   We recognize that direct solvers and machine learning-based methods each have their pros and cons, suitable for different types of problems. Therefore, in our future work, we plan to take your advice and attempt to integrate these two methods. This integration will involve using an automatic discriminator to identify the most suitable solving method for a given problem-solving task, combining the strengths of both approaches, and achieving efficient solutions for a broader range of mixed-integer programming problems.
> > > > > >
> > > > > >
> > > > > >
> > > > > >
> > > > > >
> > > > > >
> > > > > >
> > > > > > $\textbf{References}:$
> > > > > >
> > > > > > [1] Sonnerat N, Wang P, Ktena I, et al. Learning a large neighborhood search algorithm for mixed integer programs[J]. arXiv preprint arXiv:2107.10201, 2021.
> > > > > >
> > > > > > [2] Ye H, Xu H, Wang H, et al. GNN&GBDT-Guided Fast Optimizing Framework for Large-scale Integer Programming[J]. 2023.
> > > > > >
> > > > > > [3] Song J, Yue Y, Dilkina B. A general large neighborhood search framework for solving integer linear programs[J]. Advances in Neural Information Processing Systems, 2020, 33: 20012-20023.
> > > > > >
> > > > > > [4] Wu Y, Song W, Cao Z, et al. Learning large neighborhood search policy for integer programming[J]. Advances in Neural Information Processing Systems, 2021, 34: 30075-30087.
> > > > > >
> > > > > > [5] Ding J Y, Zhang C, Shen L, et al. Accelerating primal solution findings for mixed integer programs based on solution prediction[C]//Proceedings of the aaai conference on artificial intelligence. 2020, 34(02): 1452-1459.
> > > > > >
> > > > > > [6] Gasse M, Chételat D, Ferroni N, et al. Exact combinatorial optimization with graph convolutional neural networks[J]. Advances in neural information processing systems, 2019, 32.

---

> ### Author Response · Authors · 2023-11-22
>
> $\textbf{References:}$
>
> [1]  Nair V, Bartunov S, Gimeno F, et al. Solving mixed integer programs using neural networks[J]. arXiv preprint arXiv:2012.13349, 2020.
>
> [2] Sonnerat N, Wang P, Ktena I, et al. Learning a large neighborhood search algorithm for mixed integer programs[J]. arXiv preprint arXiv:2107.10201, 2021.
>
> [3] Zhang J, Liu C, Li X, et al. A survey for solving mixed integer programming via machine learning[J]. Neurocomputing, 2023, 519: 205-217.
>
> [4] Ye H, Xu H, Wang H, et al. GNN&GBDT-Guided Fast Optimizing Framework for Large-scale Integer Programming[J]. 2023.
>
> [5] Song J, Yue Y, Dilkina B. A general large neighborhood search framework for solving integer linear programs[J]. Advances in Neural Information Processing Systems, 2020, 33: 20012-20023.
>
> [6] Wu Y, Song W, Cao Z, et al. Learning large neighborhood search policy for integer programming[J]. Advances in Neural Information Processing Systems, 2021, 34: 30075-30087.

---

### Official Review · Reviewer_6e4z · 2023-10-28

**Soundness:** 3 good
**Presentation:** 1 poor
**Contribution:** 2 fair
**Rating:** 5
**Confidence:** 4

**Summary:**

The paper proposes a light-weight framework for large-scale MILP problems.  The framework consists four stages, and employs advanced machine/deep learning techniques to 1). find and improve solution 2). reduce the original problem into smaller subproblems 3). coordinate between subproblems. Numerical experiments are conducted on large-scale MILP instances to demonstrate the efficiency of the proposed method.

**Strengths:**

The paper proposes a framework that integrates several advanced tools from ML for MILP optimization. By breaking up the problem into subproblems, the framework only needs to be trained on small-scale datasets and exhibits less dependency on the capability of the MILP solver adopted.

**Weaknesses:**

Overall the presentation the paper is not accessible to the readers. There are a number of grammar and stylistic issues, making it difficult to understand the details of the proposed method.

I also have some concerns around other aspects of the proposed method. See questions below.

**Questions:**

1. The four stages in contribution 2 are actually part of contribution 1. Can you combine them?

2. In Section 2.1, "A feasible solution is optimal if it attains the minimum objective function value of the minimum MILP".

   What's minimum MILP here? Also it is better to mention unbounded and infeasible MILPs here.

3. How does the approach guarantee feasibility of the solution?

   The challenge of many real-life MILPs is to find a feasible solution. Most of the testing problems in the paper (all except case-study, which is unknown) seem to admit trivial feasible solution. How does your method work on problems like set partitioning?

4. It seems that the framework has more focus on the primal side and cannot help improving dual bound. Is there a way your method can certificate optimality?

5. It is recommended the authors conduct some ablation studies. For example, what if you directly feed the initial solution from *Model-based Initial Solution Prediction* to Gurobi?

6. Table 2 says "under the same optimization solution". What if two different solutions have the same objective value?

7. While the paper suggests the approach only needs to rely on small-scale solvers. The choice of variable proportion parameter $\alpha$ still lead to subproblems of large scale. Will tiny values of $\alpha$ (e.g., 0.01) give better/worse results?

**Minor typos and stylistic issues**

1. Section 2.4: node information. while on the other hand

   "." => ","

2. Figure 2 uses MIP instead of MILP. Please be consistent.

3. Figure 2 contains much information and is a bit hard to parse due to massive legends. Is it possible to move "data-drive" stage below the first three stages? The rest of space can be used to make legends look better.

4. Table 2: caption

   Comparsion => comparison

5. Page 13: where $E$ is a tensor representing the edge feautres

   feautres => features

6. Table 3: caption

   SC denots the Set Covering problem

   denots => denotes

---

> ### Author Response · Authors · 2023-11-22
>
> Thank you very much for your constructive questions. These inquiries undoubtedly contribute to further refining our paper and advancing the development of the field. Below, we will provide detailed responses to your questions.
>
> - $\textbf{Q0}:$ Overall the presentation the paper is not accessible to the readers. There are a number of grammar and stylistic issues, making it difficult to understand the details of the proposed method.
>
>   $\textbf{A0}:$ Thank you for your suggestions. We will thoroughly and attentively read through the entire manuscript, making revisions to enhance overall logical flow and expression. We will address grammar issues and make modifications to improve the manuscript. The updated PDF with these changes will be visible in the new submission.
>
> - $\textbf{Q1}:$  The four stages in contribution 2 are actually part of contribution 1. Can you combine them?
>
>   $\textbf{A1}:$ Thank you for your suggestions! We modify the innovations to the following two points:
>
>   1. We propose the first lightweight framework that solves large-scale MILPs with only small-scale training data and small-scale optimizers, introducing Problem Formulation, Model-based Initial Solution Prediction, Problem Reduction, and Data-driven Optimization to reduce the computational complexity of the model and improve the dimensionality reduction capability, respectively.
>   1. We demonstrate the effectiveness of the proposed framework in solving large-scale MILPs with small-scale optimizers and small training datasets through a comparative analysis with state-of-the-art ML-based optimization frameworks and advanced solvers, providing initial insights into efficiently solving large-scale MILPs with limited computational resources.
>
> - $\textbf{Q2}:$ In Section 2.1, "A feasible solution is optimal if it attains the minimum objective function value of the minimum MILP".  What's minimum MILP here? Also it is better to mention unbounded and infeasible MILPs here.
>
>   $\textbf{A2}:$ Thank you for your response!
>
>   1. The term "minimum MILP" here refers to a minimized Mixed-Integer Linear Programming (MILP) problem, specifically one that involves minimizing the objective function. This clarification has been incorporated into the revised PDF submitted.
>   2. MILP can be classified in various ways. Based on feasibility, it can be categorized as feasible MILP or infeasible MILP; based on constraint space, it can be categorized as bounded MILP or unbounded MILP. This paper focuses on feasible and bounded MILP. Additionally, we will provide a supplementary appendix [2, 3, 4] for a comprehensive classification and definition of MILP.

---

> ### Author Response · Authors · 2023-11-22
>
> - $\textbf{Q3}:$ How does the approach guarantee feasibility of the solution? The challenge of many real-life MILPs is to find a feasible solution. Most of the testing problems in the paper (all except case-study, which is unknown) seem to admit trivial feasible solution. How does your method work on problems like set partitioning?
>
>   $\textbf{A3}:$ Thank you for your question! In real-world Mixed-Integer Linear Programming (MILP) problems, finding feasible solutions is often challenging [5], especially when dealing with large-scale problems where solution times are difficult to accept. To address the challenge of initial solution prediction in large-scale problems, we propose a novel method.
>
>   1. $\textbf{Model-Based Initial Solution Prediction Module:}$ Classical approaches in the model-based initial solution prediction module often use Graph Convolutional Neural Networks (GCN) [6, 7]. However, these methods do not consider different correlations between neighborhoods. Ding et al. [8] introduced Graph Attention Networks (GAT) with an attention mechanism to capture correlations between nodes for better initial solution prediction. However, GAT only updates node features, neglecting the positive effects of updating edge features for neighborhood aggregation. Therefore, we further introduce Edge-Updated Graph Attention Networks (EGAT) with half convolutions layers to achieve higher computational efficiency.
>
>   2. $\textbf{Integration of Repair Strategy:}$ However, relying solely on neural networks for initial solution prediction is often unreliable. Therefore, this paper introduces a Repair strategy (as shown in Algorithm 5 in the Appendix). It automatically identifies predicted decision variables that violate constraints, utilizes a small-scale solver to obtain a feasible prediction, and finally obtains an initial feasible solution, effectively avoiding the problem of not finding an initial feasible solution.
>
>   3. $\textbf{Additional test:}$ The Set Partitioning Problem is a type of SAT problem without an objective function, making it unsuitable for MILP problem-solving. Many existing methods [1, 9, 10] are often tested on standard benchmarks with straightforward feasible solutions. To further validate the effectiveness of the proposed method in solving initial feasible solutions for complex real-world problems, we conducted tests on real-world problems SCP from MIPLIB. The results demonstrate that the proposed framework can effectively predict initial solutions for real-world problems and efficiently solve them. The table below presents experimental results. In the table, Ours-30%S represents the results obtained by the proposed framework using a restricted version of SCIP with a solution size limited to 30% of the original problem, and Ours-50%G represents the results obtained using a restricted version of Gurobi with a solution size limited to 50% of the original problem.
>
>      |               |   scpm1   |   scpn2   |
>      | :-----------: | :-------: | :-------: |
>      | **Ours-30%S** | **718.0** | **666.0** |
>      | **Ours-50%S** |   754.0   |   700.0   |
>      |   **SCIP**    |   807.0   |  19145.0  |
>      | **Ours-30%G** | **662.0** |   665.0   |
>      | **Ours-50%G** |   676.0   | **604.0** |
>      |  **Gurobi**   |   836.0   |   793.0   |
>      |   **Time**    |   2000s   |   2000s   |
>
>
> - $\textbf{Q4}:$  It seems that the framework has more focus on the primal side and cannot help improving dual bound. Is there a way your method can certificate optimality?
>
>   $\textbf{A4}:$​  Thank you for your response!
>
>   The framework proposed in this paper is fundamentally a heuristic approach. Heuristic search methods can achieve better results than state-of-the-art solvers and solution frameworks within a fixed and limited time. This approach has a clear advantage in situations with limited computational resources. However, it may suffer from the drawback of getting stuck in local optima. To overcome this issue, the paper explores the combination of neighborhood search and neighborhood crossover, leveraging prior knowledge derived from the partitioning of large-scale problems and the update of the active constraint set.
>
>   On large-scale problem solving using existing datasets, the proposed approach demonstrates better results than the commercial solver Gurobi. While branch-and-bound-based search methods can guarantee optimality with unlimited computational resources, in real-world scenarios with only finite computational resources, efficiently solving large-scale MILP problems becomes challenging. Therefore, in such situations, we ultimately opted for a heuristic framework approach.

---

> ### Author Response · Authors · 2023-11-22
>
> - $\textbf{Q5}:$ It is recommended the authors conduct some ablation studies. For example, what if you directly feed the initial solution from *Model-based Initial Solution Prediction* to Gurobi?
>
>   $\textbf{A5}:$ Thank you for your questions!
>
>   In the line of heuristic-based approximate solving algorithms, we provided additional comparisons with relevant work. The experimental results are shown in the tables below.
>
>   Firstly, we compared the initial solution generation strategy on three standard benchmark problems against the proposed method, academic solver SCIP, and commercial solver Gurobi. The results indicate that, under the same generation time, our method, aided by initial solution prediction, can obtain better initial feasible solutions using a small-scale solver compared to SCIP and Gurobi.
>
>   |              |  SC$_1$  |    SC$_2$     |   MVC$_1$    |    MVC$_2$    |   MIS$_1$    |    MIS$_2$    |
>   | :----------: | :------: | :-----------: | :----------: | :-----------: | :----------: | :-----------: |
>   | **Ours-30%** | 29962.09 |   295981.16   |   34634.96   |   346677.69   |   15275.82   |   158352.06   |
>   | **Ours-50%** | 30694.93 | **312419.34** | **33280.27** | **333192.51** | **16384.02** | **167161.47** |
>   |  **Gurobi**  | 32040.60 |   320272.92   |   33802.45   |   338198.32   |   16190.70   |   161973.05   |
>
>   Further analysis reveals that our initial solution prediction method not only produces better initial feasible solutions but also significantly improves efficiency compared to Gurobi. Our method requires only 20% of the time Gurobi needs to generate feasible solutions while achieving better solution quality.
>
>   |                 |   SC$_1$   |   SC$_2$    |  MVC$_1$   |   MVC$_2$   |  MIS$_1$   |   MIS$_2$   |
>   | :-------------: | :--------: | :---------: | :--------: | :---------: | :--------: | :---------: |
>   |  **Our-Time**   | **68.88s** | **655.00s** | **53.81s** | **495.38s** | **49.60s** | **480.74s** |
>   | **Gurobi-Time** |  336.77s   |  3399.88s   |  375.37s   |  2702.16s   |  279.23s   |  2742.89s   |
>
>   Next, regarding the decision variable reduction strategy, we compared SelectNet and confidence methods. The results are shown in the table below. "Ours-30%" represents the results obtained by reducing decision variables to 30% of the original problem size using the confidence method, and "SelectiveNet-50%" represents the results obtained by reducing decision variables to 50% using the SelectiveNet method. The results indicate that SelectiveNet, chosen in this study, has advantages in three problems and is inferior in the remaining two. Considering that SelectiveNet requires retraining the network for different reduction ratios, the confidence method, trained once, can be used for different ratios. Therefore, we ultimately chose the confidence-based selection method. Further analysis revealed that the fixed selection threshold in SelectiveNet led to conservative predictions, and in future experiments, we combined the strengths of SelectNet and confidence methods, resulting in better results. We appreciate your suggestion, and we will explore a new method that combines SelectNet and confidence to further improve initial solution prediction capabilities.
>
>   |                      |    SC$_1$    |    SC$_2$     |   MVC$_1$    |    MVC$_2$    |   MIS$_1$    |    MIS$_2$    |
>   | :------------------: | :----------: | :-----------: | :----------: | :-----------: | :----------: | :-----------: |
>   |     **Ours-30%**     |   29962.09   |   295981.16   | **34634.96** | **346677.69** | **15275.82** | **158352.06** |
>   | **SelectiveNet-30%** | **25605.41** | **261728.93** |   49993.15   |   500171.37   |   13804.92   |   145147.94   |
>   |     **Ours-50%**     |   30694.93   |   312419.34   | **33280.27** | **333192.51** | **16384.02** | **167161.47** |
>   | **SelectiveNet-50%** | **25605.41** | **261728.93** |   49993.15   |   500171.37   |   13804.92   |   145147.94   |
>
>   Finally, concerning the selection of the neural network structure in initial feasible solution prediction, we explored GNN & GBDT structures, integrated GAT structures, and integrated EGAT structures with multiple layers of half-convolutional layers. We conducted comparative tests on small-scale maximization MILP problems, where Var-30% represents predicting 70% of decision variables and solving the remaining 30% using a small-scale solver. Higher values indicate better prediction performance. The results indicate that the EGAT method chosen in this study, with multiple layers of half-convolutional structures, has a clear advantage.
>
>   |         | GNN&GBDT |  GAT   |    EGAT    |
>   | :-----: | :------: | :----: | :--------: |
>   | **30%** |  1817.3  | 1530.7 | **1872.3** |
>   | **40%** |  1928.5  | 1681.8 | **2002.3** |
>   | **50%** |  2007.0  | 1838.2 | **2067.9** |
>   | **60%** |  2036.1  | 1987.8 | **2096.5** |

---

> ### Author Response · Authors · 2023-11-22
>
> - $\textbf{Q6}:$ Table 2 says "under the same optimization solution". What if two different solutions have the same objective value?
>
>   $\textbf{A6}:$ Thank you for your response!
>
>   Regarding the comparison experiments presented in Table 2 on page 9 of the main PDF, it provides a performance comparison of solving time under the same optimization objective. The approach involves setting a target value in the "Target" column of the last row, representing the minimum objective function value to be achieved. The values in the table for different solution approaches represent the time required to obtain a solution with a performance not lower than the set target value within the specified time. Similar analytical methods have been adopted in recent years [1].
>
> - $\textbf{Q7}:$ While the paper suggests the approach only needs to rely on small-scale solvers. The choice of variable proportion parameter still lead to subproblems of large scale. Will tiny values of (e.g., 0.01) give better/worse results?
>
>   $\textbf{A7}:$ Thank you for providing additional information!
>
>   1. The key innovation in this paper lies in the dimensionality reduction of decision variables and constraints. The choice of the decision variable dimensionality reduction parameter Alpha (which is initially set empirically at around 30% [1]) remains fixed throughout the process and does not change, ensuring that it remains small-scale and does not become large-scale. For constraint dimensionality reduction, to mitigate the empirical nature, a progressive strategy is employed. Initially, a significant number of constraint conditions are retained. As the optimization progresses, redundant constraints are automatically identified based on the current solution, and the number of constraints is gradually reduced by removing these redundancies. Experimental results (see table) demonstrate that for the majority of baseline tests, large-scale problems achieve a reduction of decision variables to 30% of the original and a reduction of constraints to 20% of the original. The automation of Alpha selection is a target for our next research steps, and we plan to explore learning-based algorithms for the automatic determination of Alpha values.
>   2. Alpha is an empirical parameter, and its variation results in a roughly unimodal function. The effective utilization of small-scale solvers for solving large-scale problems is observed when Alpha falls within a suitable range. Too small a value for Alpha has been empirically found to be ineffective for achieving optimal results.
>
> - $\textbf{Q8}:$ Minor typos and stylistic issues.
>
>   $\textbf{A8}:$ Thank you for your comments! Mistakes will be corrected in accordance with the review comments, as detailed in the newly submitted PDF.
>
>
>
> If you have any more questions or if there's anything specific you'd like assistance with, feel free to let me know!
>
> $\textbf{References:}$
>
>  [1] Ye H, Xu H, Wang H, et al. GNN&GBDT-Guided Fast Optimizing Framework for Large-scale Integer Programming[J]. 2023.
>
> [2] Fischetti M, Ljubić I, Monaci M, et al. On the use of intersection cuts for bilevel optimization[J]. Mathematical Programming, 2018, 172(1-2): 77-103.
>
> [3] Argüello A, Rider M J. A Discrete Time Domain-Based MILP Framework for Control Parameter Tuning[J]. IEEE Systems Journal, 2020, 15(3): 3462-3469.
>
> [4] Earl M G, D'andrea R. Iterative MILP methods for vehicle-control problems[J]. IEEE Transactions on Robotics, 2005, 21(6): 1158-1167.
>
> [5] Fischetti M, Glover F, Lodi A. The feasibility pump[J]. Mathematical Programming, 2005, 104: 91-104.
>
> [6] Nair V, Bartunov S, Gimeno F, et al. Solving mixed integer programs using neural networks[J]. arXiv preprint arXiv:2012.13349, 2020.
>
> [7] Gasse M, Chételat D, Ferroni N, et al. Exact combinatorial optimization with graph convolutional neural networks[J]. Advances in neural information processing systems, 2019, 32.
>
> [8] Ding J Y, Zhang C, Shen L, et al. Accelerating primal solution findings for mixed integer programs based on solution prediction[C]//Proceedings of the aaai conference on artificial intelligence. 2020, 34(02): 1452-1459.
>
> [9] Song J, Yue Y, Dilkina B. A general large neighborhood search framework for solving integer linear programs[J]. Advances in Neural Information Processing Systems, 2020, 33: 20012-20023.
>
> [10] Wu Y, Song W, Cao Z, et al. Learning large neighborhood search policy for integer programming[J]. Advances in Neural Information Processing Systems, 2021, 34: 30075-30087.

---

> ### Comment · Reviewer_6e4z · 2023-11-22
> **Thank you for your response**
>
> I appreciate the efforts the authors made in addressing my questions. It resolved some of my concerns, while I believe there are still two major limitations.
>
> 1. Finding a feasible solution using the REPAIR Algorithm
>
>    I'm concerned that REPAIR Algorithm may not work well for real MILP problems. To me it's not obvious that the proposed strategy, which looks like some fix/propogate/repair procedure from MILP, can efficiently restore a feasible solution. The algorithm loops through each constraint sequentially, but it's still likely to stop without a feasible solution.
>
>    That being said, I think its performance might be improved by introducing LP-based heuristics (e.g. diving) from MILP literature.
>
> 2. Restrictions on $\alpha$
>
>    It looks to me that $\alpha$ has to be $\mathcal{O}(1)$ (e.g. > 0.1) to give good performance. This means the subproblem is essentially on the same order of magnitude as the original problem. For practical efficiency $\alpha$ is expected to be smaller.
>
> I again thank the authors for their response and agree that the proposed method has its potential. But there are still several unaddressed aspects I think deserves a more careful treatment. Overall I raise my score to 5.

---

> ### Author Response · Authors · 2023-11-22
> **Thank you very much for your response**
>
> Thank you very much for your response and the affirmation of our work. Your encouragement has given us great confidence. Below, I will address each of the limitations you pointed out in our work.
>
> - $\textbf{L1}$: Finding a feasible solution using the REPAIR Algorithm. I'm concerned that REPAIR Algorithm may not work well for real MILP problems. To me it's not obvious that the proposed strategy, which looks like some fix/propogate/repair procedure from MILP, can efficiently restore a feasible solution. The algorithm loops through each constraint sequentially, but it's still likely to stop without a feasible solution. That being said, I think its performance might be improved by introducing LP-based heuristics (e.g. diving) from MILP literature.
>
>   $\textbf{A1}$: Thank you for your insights! Machine learning-based algorithms for solving mixed-integer programming (MILP) problems can be broadly categorized into two types[1]: exact solution algorithms based on branch-and-bound and heuristic-based approximation algorithms. There is extensive research on branch-and-bound algorithms, covering aspects such as variable selection[2], node selection[3], and cutting plane selection[4]. Admittedly, exact solution algorithms based on branch-and-bound, like Neural Diving[5], exhibit excellent performance in many problems, particularly in cases with sparse feasible solutions, and can achieve optimal solutions given unlimited computation time and resources.
>
>   However, for real-world large-scale MILP problems with millions of decision variables and constraints, such as the optimization scheduling problem in the internet domain we addressed in our experiments (see Table 1 and Table 2), there are strict constraints on computational resources and time in practical scenarios (e.g., limited available computational resources and calculation time for optimization ). Existing exact solution algorithms based on branch-and-bound struggle to provide satisfactory solutions within these constraints, while often failing to find feasible solutions within a limited time. Consequently, our research focuses on heuristic-based approximation algorithms tailored to these practical applications.
>
>   While heuristic methods may not guarantee to finding feasible or optimal solutions, our approach, leveraging FENNEL-based problem partitioning, EGAT-based initial solution prediction, and the Repair strategy, consistently produces promising feasible solutions on standard benchmark problems and several real-world problems. In the realm of heuristic-based approximation algorithms for solving MILP problems, our framework demonstrates clear advantages. Thanks to efficient problem dimensionality reduction and rapid iterative improvement algorithms, our framework outperforms  branch-and-bound algorithm-based baseline solvers SCIP and Gurobi, as well as the state-of-the-art machine learning-based heuristic framework GNN&GBDT, even on problems with millions or tens of millions of variables and constraints, and does so on a single machine within an acceptable time range.
>
>
>   It is evident that both exact solution algorithms based on branch-and-bound and heuristic-based approximation algorithms have their merits and are suited to different types of problems. Therefore, we will carefully heed your advice and attempt to integrate both approaches to further improve the performance of the optimization architecture in future research. We will consider using an automatic discriminator to determine the suitable solution method for a given task, combining the strengths of both methods to efficiently solve a broader range of mixed-integer programming problems.

---

> ### Author Response · Authors · 2023-11-22
> **Thank you very much for your response**
>
> - $\textbf{L2}$: Estrictions on $\alpha$. It looks to me that has to be $O(1)$ (e.g. > 0.1) to give good performance. This means the subproblem is essentially on the same order of magnitude as the original problem. For practical efficiency is expected to be smaller.
>
> - $\textbf{A2}$:  Thank you for your insights! Your understanding is absolutely correct. Regarding the chosen dimension reduction ratio \( $\alpha$ \), it can indeed only be at an \( $\mathcal{O}(1)$ \) level, and achieving a final subproblem significantly smaller than the original problem is not feasible. However, this has crucial practical implications.
>
>   For mixed-integer programming problems, people typically resort to solvers. The scalability of solvers depends on both internal algorithmic settings and the user's machine configuration. In a given solver environment and computational resource constraints, individuals can only solve problems within a specific size limit, for example, those with \( $k$ \) decision variables. Previous research on machine learning-based solutions for large-scale mixed-integer programming problems also often required solvers of the same size as the problem being tackled[6,7].
>
>   In our work, we introduced a dimension reduction approach that simultaneously operates on the decision variable and constraint levels. In our test problems, when the number of decision variables reduces to 30% and the number of constraints reduces to 20% of the original problem, our framework can still generate high-quality feasible solutions. This implies that within a given solver environment and computational resource constraints, we can surpass the limits imposed by solving problems of specific sizes (e.g., with \( $k$ \) decision variables) and efficiently solve larger-scale mixed-integer programming problems (e.g., with \( $2k$ \) or \( $3k$ \) decision variables). Therefore, proposing the dimension reduction ratio \( $\alpha$ \) not only improves the efficiency of problem-solving but also serves as inspiration for overcoming physical resource constraints in solving even larger-scale problems.
>
> Thanks again for your response! If you have any further questions or if there's anything else you would like clarification on, feel free to let me know!
>
> $\textbf{References}:$
>
> [1] Zhang J, Liu C, Li X, et al. A survey for solving mixed integer programming via machine learning[J]. Neurocomputing, 2023, 519: 205-217.
>
> [2] Nair V, Bartunov S, Gimeno F, et al. Solving mixed integer programs using neural networks[J]. arXiv preprint arXiv:2012.13349, 2020.
>
> [3] Song J, Lanka R, Yue Y, et al. Co-training for policy learning[C] Uncertainty in Artificial Intelligence. PMLR, 2020: 1191-1201.
>
> [4] Huang Z, Wang K, Liu F, et al. Learning to select cuts for efficient mixed-integer programming[J]. Pattern Recognition, 2022, 123: 108353.
>
> [5] Nair V, Bartunov S, Gimeno F, et al. Solving mixed integer programs using neural networks[J]. arXiv preprint arXiv:2012.13349, 2020.
>
> [6] Sonnerat N, Wang P, Ktena I, et al. Learning a large neighborhood search algorithm for mixed integer programs[J]. arXiv preprint arXiv:2107.10201, 2021.
>
> [7] Wu Y, Song W, Cao Z, et al. Learning large neighborhood search policy for integer programming[J]. Advances in Neural Information Processing Systems, 2021, 34: 30075-30087.

---

> ### Comment · Reviewer_6e4z · 2023-12-04
> **Thanks for the response**
>
> I thank the authors for the efforts addressing my concerns. I believe that the methodology this paper proposes has its potential. However, there are unaddressed issues that are worth more careful consideration, including
>
> 1. More thorough benchmarking and ablation study
> 2. Further reduction in subproblem size
>
> Overall I keep my current score.

---

### Official Review · Reviewer_LzLN · 2023-10-28

**Soundness:** 3 good
**Presentation:** 4 excellent
**Contribution:** 3 good
**Rating:** 6
**Confidence:** 3

**Summary:**

This paper proposes Light-MILPopt, a lightweight large-scale optimization framework that only uses a small-scale optimizer and small training dataset to solve large-scale MILPs. Experiments show that it outperforms both the SOTA ML-based approaches and modern MILP solvers like Gurobi and SCIP.

**Strengths:**

1. Clear writing. the paper is very clearly structured and easy to go through flow.
2. Impressive results on large-scale MILPs. The experiments on large scale MILPs (significantly larger than those used in previous research) is impressive. The results suggest that Light-MILPopt significantly improves both the efficiency and effectiveness of solving MILPs.
3. Practical motivation. Light-MILPopt achieves such improvement with only a small-scale optimizer and small training datasets. This setting is practical for real-world applications.

**Weaknesses:**

1. Lack of comparison with previous work. From my perspective, the impressive results in this article are based on the results and approaches in many previous studies. However, both the detailed related work and the empirical comparisons to the previous research are missing. For example, previous research has pointed out that we can use a fixed threshold to replace the complex selective networks, which might motivate the variable reduction approach in this paper.
2. More analysis and insights on the proposed modules are encouraged. In this paper, the authors proposed multiple techniques to improve the performance of different modules. Though I believe these techniques are all effective, can you give more clues about the motivations of them? For example, What is the logic behind your design of the EGAT with half-convolutions? Is it the only choice for this module?
3. The nomenclature "large-scale/small-scale optimizer" appears unconventional. It might be more precise to employ terminology such as "advanced/modern MILP solver" and "lightweight solver".

Overall, I appreciate the results in this paper. If the author can provide more details and insights, I am willing to further improve my scores.

**Questions:**

See weaknesses above.

---

> ### Author Response · Authors · 2023-11-21
>
> We appreciate your attention and recognition of our work, particularly your acknowledgment of the practical real-world application value in using lightweight solvers and small-scale training data to solve large-scale problems. Additionally, we extend our gratitude for the constructive questions you have raised, which undoubtedly contribute to the further refinement of our paper and the advancement of the field. Below, we provide detailed responses to your questions.
>
> - $\textbf{Q1:}$ Lack of comparison with previous work. From my perspective, the impressive results in this article are based on the results and approaches in many previous studies. However, both the detailed related work and the empirical comparisons to the previous research are missing. For example, previous research has pointed out that we can use a fixed threshold to replace the complex selective networks, which might motivate the variable reduction approach in this paper.
>
>   $\textbf{A1:}$ Thank you for your inquiries!
>
>   1. Currently, optimization methods for mixed-integer programming problems based on machine learning can be broadly categorized into two classes [1]: exact solving algorithms based on branch-and-bound and heuristic-based approximate solving algorithms. Exact solving algorithms based on branch-and-bound [2,3,4], as the problem size increases, experience an exponential increase in complexity, resulting in low efficiency when tackling large-scale real-world mixed-integer programming problems [5]. Therefore, we have opted for the heuristic-based approximate solving algorithm approach.
>
>   2. In the heuristic-based approximate solving algorithm approach, we have provided additional comparisons with related work, including more detailed comparisons in **variable reduction methods**, network structure selection, and initial solution generation strategies. The experimental results are shown in the table below.
>
>      First is the choice of decision variable reduction strategy, where we compare two methods: SelectNet and confidence-based reduction. The experimental results are presented in the table below. Ours-30% represents the initial feasible solution results obtained by reducing decision variables to 30% of the original problem using confidence-based reduction, while SelectiveNet-50% represents the results obtained by reducing decision variables to 50% of the original problem using SelectiveNet. The results indicate that SelectiveNet selected in this paper has an advantage in one of the three problems and is noticeably inferior in the remaining two problems compared to the current method. Considering that SelectiveNet requires retraining the network for different reduction ratios, whereas the confidence-based method requires training only once and can be used for different reduction ratios, we ultimately chose the confidence-based selection method. Furthermore, further analysis revealed that the fixed selection threshold in SelectiveNet led to conservative model predictions. In subsequent experiments, we combined the strengths of SelectNet and confidence-based methods, resulting in better overall results. We appreciate your suggestion, and we plan to explore new methods that integrate SelectNet and confidence in future work to further improve the prediction capabilities of initial solutions.
>
>      |                      |    SC$_1$    |    SC$_2$     |   MVC$_1$    |    MVC$_2$    |   MIS$_1$    |    MIS$_2$    |
>      | :------------------: | :----------: | :-----------: | :----------: | :-----------: | :----------: | :-----------: |
>      |     **Ours-30%**     |   29962.09   |   295981.16   | **34634.96** | **346677.69** | **15275.82** | **158352.06** |
>      | **SelectiveNet-30%** | **25605.41** | **261728.93** |   49993.15   |   500171.37   |   13804.92   |   145147.94   |
>      |     **Ours-50%**     |   30694.93   |   312419.34   | **33280.27** | **333192.51** | **16384.02** | **167161.47** |
>      | **SelectiveNet-50%** | **25605.41** | **261728.93** |   49993.15   |   500171.37   |   13804.92   |   145147.94   |

---

> ### Author Response · Authors · 2023-11-21
>
> Next is the selection of network structure in initial feasible solution prediction. In our earlier exploration, we tried the GNN&GBDT structure, the integrated GAT structure, and the integrated EGAT structure with multiple layers of half convolutions layers. We conducted comparative tests on small-scale maximization MILP problems, where Var-30% represents predicting 70% of decision variables, and the remaining 30% are solved using a small-scale solver. Higher values indicate better predictive performance. The results show that the current method, which adopts the EGAT method with multiple layers of half convolutions, has a clear advantage.
>
> |             | GNN&GBDT |  GAT   |    EGAT    |
> | :---------: | :------: | :----: | :--------: |
> | **Var-30%** |  1817.3  | 1530.7 | **1872.3** |
> | **Var-40%** |  1928.5  | 1681.8 | **2002.3** |
> | **Var-50%** |  2007.0  | 1838.2 | **2067.9** |
> | **Var-60%** |  2036.1  | 1987.8 | **2096.5** |
>
> Finally, we compare the initial solution generation capabilities of the proposed method with academic solver SCIP and commercial solver Gurobi on three standard benchmark problems. The results show that, under the same generation time, the proposed method, aided by initial solution prediction, can obtain better initial feasible solutions than SCIP and Gurobi.
>
> |              |    SC$_1$    |    SC$_2$     |   MVC$_1$    |    MVC$_2$    |   MIS$_1$    |    MIS$_2$    |
> | :----------: | :----------: | :-----------: | :----------: | :-----------: | :----------: | :-----------: |
> | **Ours-30%** | **29962.09** | **295981.16** |   34634.96   |   346677.69   |   15275.82   |   158352.06   |
> | **Ours-50%** |   30694.93   |   312419.34   | **33280.27** | **333192.51** | **16384.02** | **167161.47** |
> |  **Gurobi**  |   32040.60   |   320272.92   |   33802.45   |   338198.32   |   16190.70   |   161973.05   |
>
> Further analysis reveals that our initial solution prediction method not only obtains better initial feasible solutions but also significantly improves efficiency compared to Gurobi. It requires only 20% of the time Gurobi needs to generate feasible solutions and achieves better-quality initial feasible solutions.
>
> |                 |   SC$_1$   |   SC$_2$    |  MVC$_1$   |   MVC$_2$   |  MIS$_1$   |   MIS$_2$   |
> | :-------------: | :--------: | :---------: | :--------: | :---------: | :--------: | :---------: |
> |  **Our-Time**   | **68.88s** | **655.00s** | **53.81s** | **495.38s** | **49.60s** | **480.74s** |
> | **Gurobi-Time** |  336.77s   |  3399.88s   |  375.37s   |  2702.16s   |  279.23s   |  2742.89s   |

---

> ### Author Response · Authors · 2023-11-21
>
> - $\textbf{Q2:}$ More analysis and insights on the proposed modules are encouraged. In this paper, the authors proposed multiple techniques to improve the performance of different modules. Though I believe these techniques are all effective, can you give more clues about the motivations of them? For example, What is the logic behind your design of the EGAT with half-convolutions? Is it the only choice for this module?
>
>   $\textbf{A2:}$ Thank you for your questions!
>
>   1. In the real world, many MILP problems are large-scale and homogeneous. Traditional methods face challenges such as high complexity and cold starts when solving such problems. Therefore, we aimed to develop a framework, Light-MILPopt, that utilizes only small-scale training data to automatically learn the problem distribution for predicting initial feasible solutions. It then employs a lightweight solver to iteratively improve the current solution for efficiently solving large-scale mixed-integer programming problems. The key to achieving efficient problem solving lies in the accurate prediction of initial feasible solutions and effective problem dimensionality reduction. The model-based initial solution prediction module uses the EGAT with half convolutions layers for learning, and the problem dimensionality reduction module utilizes the confidence algorithm and KNN algorithm for decision variable and constraint reduction, respectively. Due to constraints in the paper length, the reasons and insights behind the method selection were not detailed in the paper.
>
>   2. (1) In the model-based initial solution prediction module, classical methods often use graph convolutional neural networks [2,6]. However, these methods do not consider different correlations between neighborhoods. Therefore, Ding et al. [7] introduced the GAT with an attention mechanism to capture correlations between points for better initial solution prediction. However, this GAT only updates node features and ignores the positive impact of edge feature updates on neighborhood aggregation. Therefore, we further introduced the EGAT with an edge update mechanism, combined with half convolutions layers for higher computational efficiency. We also compared the above methods and prediction methods based on GNN and GBDT in the preliminary exploration [5], and the experimental results were shown in Q2, indicating a clear advantage of the selected method. (2) In the dimensionality reduction module, for decision variable reduction, common methods include SelectNet and the confidence-based method, as shown in Q2. It can be seen that the confidence-based method requires only one training session to obtain results comparable to SelectNet at different reduction ratios, demonstrating significant efficiency advantages. For constraint reduction, common redundancy constraint identification methods are often based on mathematical approaches [8,9,10], which tend to be slow and less effective for large-scale problems, as demonstrated in our preliminary exploration. Therefore, based on the knowledge of the current optimized solution, we proposed a redundancy constraint identification method based on KNN. The experiments demonstrated that, combined with a progressive dimensionality reduction strategy, we can retain only 20% of the constraint conditions to obtain feasible optimized solutions for the problem.
>
>   3. Through the above analysis and relevant experimental results, each module currently possesses certain technical advantages. The coordinated efforts of these modules (see Tables 1 and 2 in the main text) achieve the effect of reducing complexity and improving solution efficiency. Ultimately, this allows for the effective solution of large-scale mixed-integer programming problems using only a lightweight solver and small-scale training data.
>
> - $\textbf{Q3:}$ The nomenclature "large-scale/small-scale optimizer" appears unconventional. It might be more precise to employ terminology such as "advanced/modern MILP solver" and "lightweight solver".
>
>   $\textbf{A3:}$ Thank you for your suggestion! We will modify the term "small-scale solver" to "lightweight solver" in the main text, as it is indeed more accurate.

---

> ### Author Response · Authors · 2023-11-21
>
> $\textbf{References:}$
>
> [1] Zhang J, Liu C, Li X, et al. A survey for solving mixed integer programming via machine learning[J]. Neurocomputing, 2023, 519: 205-217.
>
> [2] Nair V, Bartunov S, Gimeno F, et al. Solving mixed integer programs using neural networks[J]. arXiv preprint arXiv:2012.13349, 2020.
>
> [3] Song J, Lanka R, Yue Y, et al. Co-training for policy learning[C] Uncertainty in Artificial Intelligence. PMLR, 2020: 1191-1201.
>
> [4] Huang Z, Wang K, Liu F, et al. Learning to select cuts for efficient mixed-integer programming[J]. Pattern Recognition, 2022, 123: 108353.
>
> [5] Ye H, Xu H, Wang H, et al. GNN&GBDT-Guided Fast Optimizing Framework for Large-scale Integer Programming[J]. 2023.
>
> [6] Gasse M, Chételat D, Ferroni N, et al. Exact combinatorial optimization with graph convolutional neural networks[J]. Advances in neural information processing systems, 2019, 32.
>
> [7] Ding J Y, Zhang C, Shen L, et al. Accelerating primal solution findings for mixed integer programs based on solution prediction[C]//Proceedings of the aaai conference on artificial intelligence. 2020, 34(02): 1452-1459.
>
> [8] Paulraj S, Chellappan C, Natesan T R. A heuristic approach for identification of redundant constraints in linear programming models[J]. International Journal of computer mathematics, 2006, 83(8-9): 675-683.
>
> [9] Estinmgsih Y, Tjahjana R H. Some methods for identifying redundant constraints in linear programming[C] Journal of Physics: Conference Series. IOP Publishing, 2019, 1321(2): 022073.
>
> [10] Sumathi P, Paulraj S. Identification of redundant constraints in large scale linear programming problems with minimal computational effort[J]. Applied Mathematical Sciences, 2013, 7(80): 3963-3974.

---

> > ### Comment · Reviewer_LzLN · 2023-11-23
> >
> > Thank you for your response. I highly suggest incorporating the content of your reply into the discussion section of your paper. I would like to keep my current score (6).

---

> > > ### Author Response · Authors · 2023-11-23
> > > **Thank you very much for your response.**
> > >
> > > Dear Reviewer,
> > >
> > > Thank you very much for your response. We have incorporated the additional experiments and discussions into the appendix of the latest submitted PDF. Once again, we appreciate your acknowledgment of our work, and your feedback undoubtedly contributes to the ongoing improvement of our paper.
> > >
> > > Best regards.

---

### Official Review · Reviewer_tDoY · 2023-10-29

**Soundness:** 3 good
**Presentation:** 3 good
**Contribution:** 3 good
**Rating:** 6
**Confidence:** 4

**Summary:**

Problem: this paper studies the problem of using machine learning-based approaches to solve large-scale mixed integer linear programs.

Framework: since the MILP problems can be represented as bipartite graphs, the authors use the FENNEL graph partition algorithm to split the original problem into small sub-problems with low correlations. Then use Edge Aggregated Graph Attention Network and Multi-Layer Perceptron to predict the initial predicted solutions of the small-scale MILP and concatenate them to obtain the initial predicted solutions of the original large-scale MILP. With the initial solution, they confidence threshold method to reduce variable dimension, and use KNN to predict the active constraint set. Then they use Neighborhood set updating, active constraint set updating, and the REPAIR algorithm to iteratively improve solutions.

**Strengths:**

- The paper is clearly structured and well-written.
- From the numerical experiments, the proposed method can obtain better objectives within a limited time than other benchmarks.
- The proposed framework requires less computational resources to train than other benchmarks.

**Weaknesses:**

I merge this with the Questions section.

**Questions:**

- At the end of subsection 3.2, you mentioned “the initial predicted solutions of the split small-scale MILP can be concatenated to obtain the initial predicted solutions of the original large-scale MILP.” While concatenating the multiple predicted solutions of small-scale problems, it is possible to have an initial solution that is infeasible to the large-scale problem, which part of your framework can handle this?
- Is it possible to make your framework more flexible in the sense that the number of variables/constraints can be (slightly) different from your training data?

---

> ### Author Response · Authors · 2023-11-21
>
> We appreciate your attention and recognition of our work, particularly your high evaluation of the proposed framework and experimental results. Furthermore, we would like to express gratitude for the constructive questions you have raised, as they undoubtedly contribute to the refinement of our paper. Below, we provide detailed responses to your queries.
>
> - $\textbf{Q1:}$ While concatenating the multiple predicted solutions of small-scale problems, it is possible to have an initial solution that is infeasible to the large-scale problem, which part of your framework can handle this?
>
>   $\textbf{A1:}$ Thank you for your inquiries! Initial solution prediction based on machine learning often utilizes Graph Convolutional Networks (GNNs) to predict the initial solutions for the entire large-scale MILP [1,2,3,4]. However, this approach encounters two significant challenges. On one hand, as the problem scales up, the required storage resources, especially GPU memory, steadily increase, making it challenging to address large-scale or even extremely large-scale problems. On the other hand, GNNs achieve initial solution prediction by learning the distribution mapping from isomorphic mixed-integer problems to optimal solutions. Yet, in more complex problems, the predicted solutions obtained through such methods are often infeasible.
>
>   To overcome the limitations of existing methods, we propose a novel initial solution prediction strategy. To address the first challenge, we introduce the FENNEL graph partitioning algorithm to decompose large-scale MILP problems into several smaller subproblems. We predict the optimal solution for each subproblem and concatenate them to obtain the initial feasible solution for the complete problem, ensuring that both training and inference are performed only on smaller-scale problems. Regarding the second challenge, previous work attempted to predict the probability of each variable being set to 1. Subsequently, for parameters $(k_0, k_1)$, the method fixed the top $k_0$ probabilities to 1 and the bottom $k_1$ probabilities to 0 [5]. However, this approach requires setting different hyperparameters for different problems. Therefore, for mixed-integer programming problems, we propose using the Repair algorithm to rectify the current solution and obtain a feasible one. This algorithm identifies illegal constraints by canceling the predictions for certain decision variables and uses a small-scale solver to initialize the canceled decision variables. Finally, a feasible solution is obtained through this process. Refer to Appendix Algorithm 5 for detailed information on the Repair algorithm.

---

> > ### Author Response · Authors · 2023-11-21
> >
> > $\textbf{References:}$
> >
> > [1] Gasse M, Chételat D, Ferroni N, et al. Exact combinatorial optimization with graph convolutional neural networks[J]. Advances in neural information processing systems, 2019, 32.
> >
> > [2] Nair V, Bartunov S, Gimeno F, et al. Solving mixed integer programs using neural networks[J]. arXiv preprint arXiv:2012.13349, 2020.
> >
> > [3] Ding J Y, Zhang C, Shen L, et al. Accelerating primal solution findings for mixed integer programs based on solution prediction[C]//Proceedings of the aaai conference on artificial intelligence. 2020, 34(02): 1452-1459.
> >
> > [4] Ye H, Xu H, Wang H, et al. GNN&GBDT-Guided Fast Optimizing Framework for Large-scale Integer Programming[J]. 2023.
> >
> > [5] Han Q, Yang L, Chen Q, et al. A GNN-Guided Predict-and-Search Framework for Mixed-Integer Linear Programming[C]//The Eleventh International Conference on Learning Representations. 2022.

---

> ### Author Response · Authors · 2023-11-21
>
> -
>
> - $\textbf{Q2:}$ Is it possible to make your framework more flexible in the sense that the number of variables/constraints can be (slightly) different from your training data?
>
>   $\textbf{A2:}$ Thank you for your inquiries!
>
>   1. The proposed solution method for solving large-scale MILPs based on EGAT with half convolutions structure is notable for its use of the FENNEL-based problem partitioning strategy. This strategy decomposes large-scale MILPs into several smaller subproblems, predicting the optimal solution for each subproblem and concatenating them to obtain the initial feasible solution for the complete problem. This approach allows training and inference to be performed exclusively on smaller-scale problems. Therefore, for specific new optimization problems, it is not constrained by the decision variable and constraint quantities of existing training datasets, demonstrating robust generalization capabilities across different scales of new optimization problems. For instance, while the training dataset comprises small-scale data with decision variables and constraints in the order of tens of thousands, the testing dataset includes problems with decision variables and constraints ranging from hundreds of thousands to millions, as shown in Table 1 and Table 2 in the main text. The latest experiments extend the generalization capabilities of the proposed Light-MILPopt method, trained on small-scale data with decision variables and constraints in the order of tens of thousands, to solve problems with decision variables and constraints in the order of tens of millions. The problem scales are illustrated in the table below, where SC represents Set Covering, MVC represents Minimum Vertex Cover, and MIS represents Maximum Independent Set.
>
>      |                           |  SC$_3$  | MVC$_3$  | MIS$_3$  |
>      | :-----------------------: | :------: | :------: | :------: |
>      |  **Number of Variables**  | 20000000 | 10000000 | 10000000 |
>      | **Number of Constraints** | 20000000 | 30000000 | 30000000 |
>
>      The solution results are presented in the table below, where Ours-30%S and Ours-50% represent the results of the proposed framework using only 30% or 50% of the original problem scale for lightweight small-scale SCIP solving. SCIP indicates the results obtained directly using the baseline solver SCIP. It is evident that the proposed Light-MILPopt method maintains a significant advantage over the baseline solver, even when dealing with problem scales in the order of tens of millions for both decision variables and constraints.
>
>      |               |     SC$_3$     |    MVC$_3$     |    MIS$_3$     |
>      | :-----------: | :------------: | :------------: | :------------: |
>      | **Ours-30%S** | **1672097.50** |   2731152.61   |   2256644.32   |
>      | **Ours-50%S** |   2889696.49   | **2696953.27** | **2299950.04** |
>      |   **SCIP**    |   9190301.09   |   4909317.99   |    90750.01    |
>      |   **Time**    |     80000s     |     80000s     |     80000s     |
>
>   2. The proposed method also demonstrates adaptability to different scales of training data and uniform learning capabilities. Even when training data vary in scale, the model can be trained consistently and coherently. We supplement the results of training using a mixture of small-scale and medium-scale training data. Specifically, Ours represents the results of training with only small-scale training data, while Mix represents the results of training with the addition of medium-scale training data on top of small-scale training data. The results indicate that, in the case of SC and MVC problems, adding medium-scale training data enhances the generalization capability for initial solution prediction in large-scale data. Meanwhile, for MIS problems, the addition of training data of different scales yields results comparable to those without further improvement, suggesting saturation in the training data for the specific problem. Thus, even when using training data containing problems of different scales, the trained model can efficiently solve large-scale problems.
>
>      |              |    SC$_1$    |    SC$_2$     |   MVC$_1$    |    MVC$_2$    |   MIS$_1$    |    MIS$_2$    |
>      | ------------ | :----------: | :-----------: | :----------: | :-----------: | :----------: | :-----------: |
>      | **Ours-30%** |   29962.09   |   295981.16   |   34634.96   |   346677.69   | **15275.82** | **158352.06** |
>      | **Mix-30%**  | **28882.44** | **290205.17** | **31788.13** | **318258.13** |   14577.61   |   152015.11   |
>      | **Ours-50%** |   30694.93   |   312419.34   |   33280.27   |   333192.51   | **16384.02** | **167161.47** |
>      | **Mix-50%**  | **29710.68** | **300873.89** | **32424.85** | **323032.42** |   16044.14   |   162911.65   |
>
>      Thus, the method proposed in this paper demonstrates strong generalization capabilities in both the training and testing phases, effectively solving real-world large-scale MILP problems.

---

### Author Response · Authors · 2023-11-23
**General response to all reviewers and the new revision**

Dear Reviewers,

We sincerely appreciate the feedback and constructive suggestions from all the reviewers. This paper introduces Light-MILPopt, the first lightweight large-scale optimization framework that utilizes a lightweight optimizer and a small training dataset to solve large-scale Mixed-Integer Linear Programs (MILPs), surpassing state-of-the-art machine learning-based optimization frameworks and advanced large-scale solvers (e.g., Gurobi, SCIP) on four large-scale benchmark MILPs and a real-world case study.

Below, we would like to highlight some new achievements and modifications made to the paper:

- We conducted tests on ultra-large-scale datasets with tens of millions of decision variables and constraints, validating the robust generalization capability of Light-MILPopt and its outstanding solving performance on ultra-large-scale problems.
- We added additional ablation experiments, including variable reduction methods, network structure selection, and initial solution generation strategies, demonstrating that each module of Light-MILPopt has certain technical advantages.
- We included tests on real-world problems from the MIPLIB dataset, specifically on the scp problem. The results indicate that the proposed framework can effectively perform initial solution predictions for real-world practical problems and achieve efficient problem-solving.

We hope these modifications further enhance our work and address the issues raised during the review process. We are open to any further guidance or discussion and are eager to provide any necessary explanations and supplementary modifications to improve our work further. In conclusion, we sincerely thank all the reviewers for their valuable feedback, which has been instrumental in refining our research.

Best regards.

---

### Meta-Review · Area_Chair_ZuP2 · 2023-12-10

**Metareview:**

This paper considers the problem of using machine learning to efficiently solve large-scale MILP problems. The overall approach is based on the principle of divide-and-conquer by exploiting the structure of the problem. Experiments demonstrate the scalability of the approach on several benchmark problems.

Having gone through the reviews, rebuttal, and read the paper myself, I think the merits of the paper outweigh some (subjective) outstanding concerns. Authors' have done a good job in responding to the reviewers' comments and addressing them.

Therefore, I recommend accepting the paper as I feel the approach is promising and will inspire new research in this problem space. I strongly encourage the authors' to make sure that the final paper reflects the discussion and address the last two review comments from Reviewer 6e4z in an objective way through additional experiments.

**Justification For Why Not Higher Score:**

There are a couple of minor things including improvement in results and generality of the approach.

**Justification For Why Not Lower Score:**

The proposed approach is very promising and will likely inspire new research in this solution space.

---

### Decision · Program_Chairs · 2024-01-16

Accept (poster)